



# WRF-GC: online coupling of WRF and GEOS-Chem for regional atmospheric chemistry modeling, Part 1: description of the one-way model (v1.0)

Haipeng Lin[1,2], Xu Feng[1], Tzung-May Fu[3,4,*], Heng Tian[1], Yaping Ma[1], Lijuan Zhang[1], Daniel J. Jacob[2], Robert M. Yantosca[2], Melissa P. Sulprizio[2], Elizabeth W. Lundgren[2], Jiawei Zhuang[2], Qiang Zhang[5], Xiao Lu[1,2], Lin Zhang[1], Lu Shen[2], Jianping Guo[6], Sebastian D. Eastham[7], and Christoph A. Keller[8]

[1]Department of Atmospheric and Oceanic Sciences, School of Physics, Peking University, Beijing, China
[2]Harvard John A. Paulson School of Engineering and Applied Sciences, Harvard University, Cambridge, MA, USA
[3]School of Environmental Science and Engineering, Southern University of Science and Technology, Shenzhen, Guangdong, China
[4]Shenzhen Institute of Sustainable Development, Southern University of Science and Technology, Shenzhen, Guangdong, China
[5]Ministry of Education Key Laboratory for Earth System Modeling, Department of Earth System Science, Tsinghua University, Beijing, China
[6]State Key Laboratory of Severe Weather & Key Laboratory of Atmospheric Chemistry of CMA, Chinese Academy of Meteorological Sciences, Beijing, China
[7]Laboratory for Aviation and the Environment, Massachusetts Institute of Technology, Cambridge, MA, USA
[8]Universities Space Research Association, Columbia, Maryland, USA

**Correspondence:** Tzung-May Fu (fuzm@sustech.edu.cn)

**Abstract.** We developed the WRF-GC model, an online coupling of the Weather Research and Forecasting (WRF) mesoscale meteorological model and the GEOS-Chem atmospheric chemistry model, for regional atmospheric chemistry and air quality modeling. Both WRF and GEOS-Chem are open-source and community-supported. WRF-GC provides regional chemistry modellers easy access to the GEOS-Chem chemical module, which is stably-configured, state-of-the-science, well-documented, traceable, benchmarked, actively developed by a large international user base, and centrally managed by a dedicated support team. At the same time, WRF-GC gives GEOS-Chem users the ability to perform high-resolution forecasts and hindcasts for any location and time of interest. WRF-GC is designed to be easy to use, massively parallel, extendable, and easy to update. The WRF-GC coupling structure allows future versions of either one of the two parent models to be immediately integrated into WRF-GC. This enables WRF-GC to stay state-of-the-science with traceability to parent model versions. Physical and chemical state variables in WRF and in GEOS-Chem are managed in distributed memory and translated between the two models by the WRF-GC Coupler at runtime. We used the WRF-GC model to simulate surface $PM_{2.5}$ concentrations over China during January 22 to 27, 2015 and compared the results to surface observations and the outcomes from a GEOS-Chem nested-grid simulation. Both models were able to reproduce the observed spatiotemporal variations of regional $PM_{2.5}$, but the WRF-GC model (r = 0.68, bias = 29%) reproduced the observed daily $PM_{2.5}$ concentrations over Eastern China better than the GEOS-Chem model did (r = 0.72, bias = 55%). This was mainly because our WRF-GC simulation, nudged with surface and upper-level meteorological observations, was able to better represent the spatiotemporal variability of the





planetary boundary layer heights over China during the simulation period. Both parent models and the WRF-GC Coupler are parallelized across computational cores and can scale to massively parallel architectures. The WRF-GC simulation was three times more efficient than the GEOS-Chem nested-grid simulation at similar resolutions and for the same number of computa-

20 tional cores, owing to the more efficient transport algorithm and the MPI-based parallelization provided by the WRF software framework. WRF-GC scales nearly perfectly up to a few hundred cores on a variety of computational platforms. Version 1.0 of the WRF-GC model supports one-way coupling only, using WRF-simulated meteorological fields to drive GEOS-Chem with no feedbacks from GEOS-Chem. The development of two-way coupling capabilities, i.e., the ability to simulate radiative and microphysical feedbacks of chemistry to meteorology, is under-way. The WRF-GC model is open-source and freely available

from http://wrf.geos-chem.org.

## 1 Introduction

Regional models of atmospheric chemistry simulate the emission, transport, chemical evolution, and removal of atmospheric constituents over a regional domain. These models are widely useful for forecasts of air quality, for impact-assessment associated with polluting activities, and for theory-validation by comparisons against observations. It is thus crucial that regional

models be frequently updated to reflect the latest scientific understandings of atmospheric processes. At the same time, the increasing demand for fine-resolution simulations requires models to adapt to massively parallel computation structures with high scalability. We present here the development of a new regional atmospheric chemistry model, WRF-GC, specifically designed to stay state-of-the-science and be computationally efficient, in order to better serve the public, inform policy makers, and advance science.

Regional atmospheric chemistry models fall into two categories: offline models and online models. Offline models (also called chemical transport models, CTMs) use archived meteorological fields, either those simulated by models alone or those assimilated with observations, to drive the transport and chemical evolution of atmospheric constituents (Baklanov et al., 2014). By eliminating the need to solve dynamical processes online, the developers of offline models can focus their efforts to solving more complex chemical processes. For example, one popular regional CTM is the GEOS-Chem model in its nested-grid

configuration (Bey et al., 2001; Wang et al., 2004; Chen et al., 2009; Zhang et al., 2015), which is driven by high-resolution assimilated meteorological data from the Goddard Earth Observation System (GEOS) of the NASA Global Modeling and Assimilation Office (GMAO). GEOS-Chem has undergone three major chemical updates in the last year. Its latest standard chemical mechanism (version 12.6.0 as of the time of this submission) includes state-of-the-science $O_x$-$NO_x$-VOC-halogen-aerosol reactions. In addition, GEOS-Chem offers a number of specialty simulations to address a variety of scientific questions, such as

simulations of $CO_2$ (Nassar et al., 2010), CO (Fisher et al., 2017), methane (Maasakkers et al., 2019), mercury (Horowitz et al., 2017; Soerensen et al., 2010), persistent organic pollutants (Friedman et al., 2013), and dicarbonyls (Fu et al., 2008, 2009; Cao et al., 2018). Another widely-used regional CTM is the Community Multiscale Air Quality Modeling System (CMAQ) (Byun and Schere, 2006), which is driven by meteorology fields simulated by the Weather Research and Forecasting model (WRF) (Skamarock et al., 2008). CMAQ has undergone three major chemical updates in the last four years. The standard chemical





mechanism of CMAQ (v5.3 as of the time of this submission) also includes updated options for $O_x$-$NO_x$-VOC-halogen-aerosol chemistry. Several other regional offline models in common use are summarized in Table 1. The chemical mechanisms in these offline models are generally updated at least once a year.

Despite their updated representation of chemical processes and relative ease of use, offline models have several key short-comings. First, the applications of some offline models are limited by the time span and resolution of the available meteoro-
logical data. In the case of the GEOS-Chem nested-grid model, its application is currently limited to $0.5° \times 0.625°$ or coarser resolution between 1979 and the present day when using the Modern-Era Retrospective analysis for Research and Applications, Version 2 (MERRA-2) dataset, or to $0.25° \times 0.3125°$ or coarser resolution between 2013 and the present day when using the GEOS-Forward Processing (GEOS-FP) dataset. The temporal interpolation of sparsely-archived meteorological data can also cause significant errors in the CTM simulations (Yu et al., 2018). Most importantly, offline models cannot simulate
meteorology-chemistry interactions due to the lack of chemical feedback to meteorology.

In contrast, online regional atmospheric chemistry models perform integrated meteorological and chemical calculations, managed through operator splitting (Baklanov et al., 2014). In this way, online models can simulate regional atmospheric chemistry at any location and time of interest, without the need for temporal interpolation of the meteorological variables. Moreover, online models have the option to include "two-way coupling" processes, i.e., the response of meteorology to gases
and aerosols via interactions with radiation and cloud processes. Many studies have demonstrated the importance of two-way interactions in accurate air quality simulations (e.g., Li et al. (2011); Ding et al. (2013); Wang et al. (2014a)). One of the most extensively used online models for regional atmospheric chemistry is the Weather Research and Forecasting model coupled with Chemistry (WRF-Chem), with options for either one-way or two-way coupling (Grell et al., 2005; Fast et al., 2006). The latest version of WRF-Chem (v4.1) includes many options for $O_x$-$NO_x$-VOC-aerosol chemistry. WRF-Chem simulates the
two-way interactions between chemistry and meteorology by taking into account the scattering and absorption of radiation by gases and aerosols, as well as the activation of aerosols as cloud condensation nuclei and ice nuclei (Fast et al., 2006; Gustafson et al., 2007; Chapman et al., 2009).

However, keeping the representation of atmospheric processes up-to-date is considerably more difficult for online models than it is for offline models. Table 1 summarizes some of the regional online models currently in use. These online models are
updated annually at best, considerably less frequent than the chemical updates to offline models. The reasons for the relatively infrequent updates to online models are threefold. First, the resources available to the development teams of online models are spread thinner, such that updating, benchmarking, validating, and documenting the many more individual components of online models are difficult to do in a timely way. Second, the modelling expertise for atmospheric physical and chemical processes resides in different communities, such that each community would often develop its own model variations without
communicating the changes back to the full model. As a result, model versions may quickly diverge, and the integrity of the full model is difficult to maintain. This is currently an issue with the WRF-Chem model, where the different optional schemes are developed by different communities and not always compatible with one another. Thirdly, the interactions between the chemical and meteorological modules are often hard-wired, such that updating either module requires considerable effort. An example of this last point is the online WRF-CMAQ model, which is a coupled implementation of the WRF model and the



CMAQ model (Wong et al., 2012; Yu et al., 2014). This implementation involved direct code modifications to WRF, which reduced the immediate applicability to updates of either parent models.

In this work, we developed a new online regional atmospheric chemistry model, WRF-GC, by coupling the WRF meteorology model with the GEOS-Chem chemistry model. Both WRF and GEOS-Chem are open-source and supported by the community. We developed WRF-GC with the following guidelines, in order to facilitate usage, maintenance, and extension of

model capability in the future:

1. The coupling structure of WRF-GC should be abstracted from the parent models and involve no hard-wired codes to either parent model, such that future updates of the parent models can be immediately incorporated into WRF-GC with ease.

2. The WRF-GC coupled model should scale from conventional computation hardware to massively parallel computation

architectures.

3. The WRF-GC coupled model should be easy to install and use, open-source, version-controlled, and well-documented.

WRF-GC provides users of WRF-Chem or other regional models access to the latest GEOS-Chem chemical module. The advantage of GEOS-Chem is that it is state-of-the-science, well-documented, traceable, benchmarked, actively developed by a large international user base, and centrally managed by a dedicated support team. At the same time, WRF-GC drives the GEOS-

Chem chemical module with online meteorological fields simulated by the WRF open-source meteorological model. WRF can be driven by initial and boundary meteorological conditions from many different assimilated datasets or climate model outputs (Skamarock et al., 2008, 2019). As such, WRF-GC allows GEOS-Chem users to perform high-resolution regional chemistry simulations in both forecast and hindcast modes at any location and time of interest.

In this Part 1 paper, we describe the development of the WRF-GC model (v1.0, doi:10.5281/zenodo.3550330) for simulation

over a single domain with one-way coupling capability. The nested domain and two-way coupling capabilities are under development and will be described in a forthcoming paper.

## 2   The parent models: WRF and GEOS-Chem

### 2.1   The WRF model

Meteorological processes and advection of atmospheric constituents in the WRF-GC coupled model are simulated by the

WRF model (version 3.9.1.1 or later versions). WRF is an open-source community numerical weather model designed for both research and operational applications (Skamarock et al., 2008, 2019). WRF currently uses the Advanced Research WRF (ARW) dynamical solver, which solves fully compressible, Eulerian non-hydrostatic equations on terrain-following, hybrid vertical coordinates. Vertical levels in WRF can be defined by the user. Horizontal grids in WRF are staggered Arakawa C-grids, which can be configured by the user using four map projections: latitude-longitude, Lambert conformal, Mercator, and polar

stereographic. WRF supports the use of multiple nested domains to simulate the interactions between large-scale dynamics and





meso-scale meteorology. WRF supports grid-, spectral-, and observational-nudging. This allows the WRF model to produce meteorological outputs that mimic assimilated meteorological fields for use in air quality hindcasts. The WRF model offers many options for land surface physics, planetary boundary layer physics, radiative transfer, cloud microphysics, and cumulus parameterization, for use in meteorological studies, real-time numerical weather prediction, idealized simulations, and data

assimilation on meso- to regional scales (Skamarock et al., 2008, 2019).

The WRF model incorporates a highly modular software framework that is portable across a range of computing platforms. WRF supports two-level domain decomposition for distributed-memory (MPI) and shared-memory (OpenMP) parallel computation. Distributed parallelism is implemented through the Runtime System Library lite (RSL-lite) module, which supports irregular domain decomposition, automatic index translation, distributed input/output, and low-level interfacing with MPI li-

braries (Michalakes et al., 1999).

## 2.2 The GEOS-Chem model

Our development of WRF-GC is made possible by a recent structural overhaul of GEOS-Chem (Long et al., 2015; Eastham et al., 2018), which enabled the use of GEOS-Chem as a self-contained chemical module within the WRF-GC model. The original GEOS-Chem CTM (before version 11.01) was structured specifically for several sets of static global or regional 3-D

grids at pre-determined horizontal and vertical resolutions (Bey et al., 2001). Parallelism for the original GEOS-Chem was implemented through OpenMP, which limited the deployment of the original GEOS-Chem to single-node hardware with large shared memory. Long et al. (2015) restructured the core processes in GEOS-Chem, including emission, chemistry, convective mixing, planetary boundary layer transport, and deposition processes, to work in modular units of atmospheric vertical columns. Information about the horizontal grids, formerly fixed at compile-time, are now passed to the GEOS-Chem chemical module

at runtime. This development enabled the use of the GEOS-Chem chemical module with any horizontal grid structure and horizontal resolution.

The new, modularized structure of the GEOS-Chem has been implemented in two types of configurations. The first type of configuration uses GEOS-Chem as the core of offline CTMs. For example, in the GEOS-Chem 'Classic' implementation (GCC), the GEOS-Chem chemical module is driven by the GEOS meteorological data and is parallelized using OpenMP.

This implementation treats the pre-defined global or regional model domain as a contiguous set of atmospheric columns, with vertical layers pre-configured to match those of the GEOS model. In essence, this configuration mimics the 'original' GEOS-Chem model before the structural overhaul by Long et al. (2015). Other grid systems can also be used with the GEOS-Chem chemical module. For example, the GEOS-Chem High Performance implementation (GCHP) (Eastham et al., 2018) calls the GEOS-Chem chemical module on the native cubed-sphere coordinates of the NASA GEOS model via a column interface

in GEOS-Chem, (`GIGC_Chunk_Run`). This column interface was built on the Earth System Modeling Framework (ESMF) (Eastham et al., 2018) and permits runtime specification of the horizontal grid parameters. The GCHP implementation uses MPI to parallelize GEOS-Chem across nodes through the Model Analysis and Prediction Layer framework (MAPL) (Suarez et al., 2007), which is a wrapper on top of ESMF specifically designed for the GMAO GEOS system.





Alternatively, GEOS-Chem can be used as a module coupled to weather models or Earth System models to perform online
chemical calculations. Using this capability, Hu et al. (2018) developed an online implementation of GEOS-Chem by coupling
it to the NASA GEOS-5 model to simulate global atmospheric chemistry. Lu et al. (2019) coupled GEOS-Chem to the Beijing
Climate Center Atmospheric General Circulation Model (BCC-AGCM). However, both the GEOS-5 model and the BCC-
AGCM are proprietary.

WRF-GC is the first implementation that couples the GEOS-Chem chemical module to an open-access high-resolution
meteorological model. We developed a modular coupler between WRF and GEOS-Chem that draws from the technology of
GCHP but does not rely on ESMF (described in section 3.2). We also made changes to GEOS-Chem to accept arbitrary vertical
discretization from WRF at runtime and to improve physical compatibility with WRF (described in section 3.2.1). These
changes have been incorporated into the mainline GEOS-Chem code. Our coupler and code modifications can be adapted in
the future to couple GEOS-Chem to other non-ESMF Earth System models.

Chemical calculations in WRF-GC v1.0 use the GEOS-Chem version 12.2.1 (doi:10.5281/zenodo.2580198). The standard
chemical mechanism in GEOS-Chem includes detailed $O_x$-$NO_x$-VOC-ozone-halogen-aerosol in the troposphere, as well as
the Unified tropospheric-stratospheric chemistry extension (UCX) (Eastham et al., 2014) for stratospheric chemistry and
stratosphere-troposphere exchange. The gas-phase mechanism in GEOS-Chem currently includes 241 chemical species and
981 reactions. Reactions and rates follow the latest recommendations from the Jet Propulsion Laboratory and the International
Union of Pure and Applied Chemistry. GEOS-Chem uses the `FlexChem` pre-processor (a wrapper for the Kinetic PreProces-
sor, KPP, Damian et al. (2002)) to configure chemical kinetics (Long et al., 2015). `FlexChem` also allows GEOS-Chem users
to easily add chemical species and reactions, and to develop custom mechanisms and diagnostics.

By default, aerosols in the GEOS-Chem chemical module are simulated as speciated bulk masses, including sulfate, nitrate,
ammonium, black carbon, primary organic aerosol (POA), secondary organic aerosol (SOA), dust, and sea salt. Detailed,
size-dependent aerosol microphysics are also available as options using the TwO-Moment Aerosol Sectional microphysics
(TOMAS) module (Kodros and Pierce, 2017) or the Advanced Particle Microphysics (APM) module (Yu and Luo, 2009).
However, these two options are not yet supported by WRF-GC v1.0. The thermodynamics of secondary inorganic aerosol are
coupled to gas-phase chemistry and computed with the ISORROPIA II module (Park et al., 2004; Fountoukis and Nenes, 2007;
Pye et al., 2009). Black carbon and POA are represented in GEOS-Chem as partially hydrophobic and partially hydrophilic,
with a conversion timescale from hydrophobic to hydrophilic of 1.2 days (Wang et al., 2014b). GEOS-Chem includes two
options to describe the production of SOA. By default, SOA are produced irreversibly using simple yields from volatile organic
precursors (Kim et al., 2015). Alternatively, SOA can be complexly produced from the aqueous reactions of oxidation products
from isoprene (Marais et al., 2016), as well as from the aging of semi-volatile and intermediate volatility POA using a volatility
basis set (VBS) scheme (Robinson et al., 2007; Pye et al., 2010). Dust aerosols are represented in 4 size bins (Fairlie et al.,
2007), while sea salt aerosols are represented in accumulation and coarse modes (Jaeglé et al., 2011).

All emissions in GEOS-Chem are configured at runtime using the Harvard-NASA Emissions Component (HEMCO) (Keller
et al., 2014). HEMCO allows users to select emission inventories from the GEOS-Chem library or add their own, apply scaling
factors, overlay and mask inventories among other operations, without having to edit or compile the code. HEMCO also has





extensions to compute emissions with meteorological dependencies, such as the emissions of biogenic species, soil NO$_x$,
lightning NO$_x$, sea salt, and dust.

GEOS-Chem calculates the convective transport of chemical species using a simple single-plume parameterization (Allen et al., 1996; Wu et al., 2007). Boundary-layer mixing is calculated using a non-local scheme that takes into account the magnitude of the atmospheric instability (Lin and McElroy, 2010). Dry deposition is based on a resistance-in-series scheme (Wesely, 1989; Wang et al., 1998). Aerosol deposition is as described in Zhang et al. (2001), with updates to account for size-
dependency for dust (Fairlie et al., 2007) and sea salt (Alexander et al., 2005; Jaeglé et al., 2011). Wet scavenging of gases and water-soluble aerosols in GEOS-Chem are as described in Liu et al. (2001) and Amos et al. (2012).

## 3   Description of the WRF-GC coupled model

### 3.1   Overview of the WRF-GC model architecture

Figure 1 gives an architectural overview of the WRF-GC coupled model. Our development of WRF-GC uses many of the
existing infrastructure in the WRF-Chem model that couples WRF to its chemistry module (Grell et al., 2005). The interactions between WRF and the chemistry components are exactly the same in WRF-GC and in WRF-Chem. Operator splitting in WRF-GC is exactly as it is in the WRF-Chem model. However, the chemistry components in the WRF-GC model are organized with greater modularity. Within WRF-GC, the WRF model and the GEOS-Chem model remain entirely intact. The WRF-GC Coupler interfacing the WRF and GEOS-Chem models is separate from both parent models and is written in a manner similar
to an application programming interface. The WRF-GC Coupler consists of interfaces with the two parent models, as well as a state conversion module and a state management module.

The WRF-GC model is initialized and driven by WRF, which sets up the simulation domain, establishes the global clock, sets the initial and boundary conditions for meteorological and chemical variables, handles input and output, and manages cross-processor communication for parallelization. Users define the domain, projection, simulation time, time steps, and physical
and dynamical options in the WRF configuration file (`namelist.input`). GEOS-Chem initialization is also managed by the WRF model through the WRF-to-chemistry interface. Chemical options, including the choice of chemical species, chemical mechanisms, emissions, and diagnostics, are defined by users in the GEOS-Chem configuration files (`input.geos`, `HEMCO_Config.rc`, and `HISTORY.rc`).

Dynamical and physical calculations are performed in WRF-GC exactly as they are in the WRF model. WRF also per-
forms the grid-scale advection of chemical species. At the beginning of each chemical time step, WRF calls the WRF-GC chemistry component through the WRF-to-Chemistry interface. Spatial parameters and the internal state of WRF are translated at runtime to GEOS-Chem by the state conversion and management modules. The GEOS-Chem chemical module then performs convective transport, dry deposition, wet scavenging, emission, boundary layer mixing, and chemistry calculations. This operator-splitting between WRF and GEOS-Chem is identical to that in WRF-Chem. Then, the GEOS-Chem internal
state is translated back to WRF, and the WRF time-stepping continues. At the end of the WRF-GC simulation, WRF outputs all meteorological and chemical variables and diagnostics in its standard format.





By design, WRF-GC supports all existing input and output functionality of the WRF model, including serial/parallel reading and writing of netCDF, HDF5, and GRIB2 datasets. This allows current WRF and WRF-Chem users to use existing data pre- and post-processing tools to prepare input data and analyze model results.

## 3.2 Details about the WRF-GC Coupler technology

### 3.2.1 Further modularization of GEOS-Chem for WRF-GC coupling

Long et al. (2015) re-structured the GEOS-Chem model into modular units of atmospheric columns. However, there were limitations in that column structure and its interface which prohibit the coupling with WRF. First, the GEOS-Chem module developed by Long et al. (2015) was hard-coded to operate on pre-defined configurations of either 72 or 47 vertical levels.

The former configuration was designed to match the native vertical levels of the GEOS model. The latter configuration was designed to match the lumped vertical levels often used by the GEOS-Chem 'Classic' model. Second, the column interface to the GEOS-Chem module as implemented in GCHP depends on the ESMF and MAPL frameworks, which WRF does not support.

We modified the GEOS-Chem module and interface to facilitate more flexible coupling with WRF and other dynamical

models. We allowed GEOS-Chem to accept the $A_p$ and $B_p$ parameters for the hybrid sigma-eta vertical grids and the local tropopause level from WRF at runtime. Stratospheric chemistry will only be calculated in GEOS-Chem above the tropopause level passed from WRF. Also, 3-D emissions (such as the injection of biomass burning plumes into the free troposphere) are interpolated in HEMCO to the WRF-GC vertical levels.

In addition, we modified the existing GCHP interface `GIGC_Chunk_Run` to remove its dependencies on ESMF and MAPL

when running in WRF-GC. We added a set of compatible error-handling and state management components to GEOS-Chem that interacts with the WRF-to-Chemistry interface, to replace the functionalities originally provided by ESMF. This removes all dependency of the WRF-GC Coupler and the GEOS-Chem column interface on external frameworks.

All of our changes adhere to the GEOS-Chem coding and documentation standards and have been fully merged into the GEOS-Chem standard source code as of version 12.0.0 (doi: 10.5281/zenodo.1343547) and are controlled with the pre-

240 processor switch `MODEL_WRF` at compile time. In the future, these changes will be maintained as part of the standard GEOS-Chem model.

### 3.2.2 Runtime processes

Similar to WRF-Chem, in WRF-GC all chemistry-related codes reside in the `chem/` sub-directory under the WRF model directory. These include the WRF-GC Coupler code, an unmodified copy of the GEOS-Chem code in the `chem/gc/` sub-

245 directory, and a set of sample GEOS-Chem configuration files in `chem/config/`. In WRF-Chem, WRF calls its interface to chemistry, `chem_driver`, which then calls each individual chemical processes. We abstracted this `chem_driver` interface by removing direct calls to chemical processes. Instead, our `chem_driver` calls the WRF-GC state conversion module





(`WRFGC_Convert_State_Mod`) and the GEOS-Chem column interface (`GIGC_Chunk_Run`) to perform chemical calculations.

The WRF-GC state conversion module includes two subroutines. The `WRFGC_Get_WRF` subroutine receives meteorological data and spatial information from WRF and translates them into GEOS-Chem formats and units. Table 2 summarizes the meteorological variables required to drive GEOS-Chem. Many meteorological variables in WRF only require a conversion of units before passing to GEOS-Chem. Some meteorological variables require physics-based diagnosis in the `WRFGC_Get_WRF` subroutine before passing to GEOS-Chem. For example, GEOS-Chem uses the convective mass flux variable to drive convec-

tive transport. This variable is calculated in the cumulus parameterization schemes in WRF but not saved. We re-diagnose the convective mass flux variable in `WRFGC_Get_WRF` using the user-selected cumulus parameterization schemes in WRF and pass it to GEOS-Chem. Horizontal grid coordinates and resolutions are passed to GEOS-Chem in the form of latitudes and longitudes at the center and edges of each grid. Vertical coordinates are passed from WRF to GEOS-Chem at runtime as described in Section 3.2.1. A second subroutine, `WRFGC_Set_WRF`, receives chemical species concentrations from GEOS-

Chem, converts the units, and saves them in the WRF chemistry variable array.

We developed the WRF-GC state management module (`GC_Stateful_Mod`) to manage the GEOS-Chem internal state in distributed memory, such that GEOS-Chem can run in the MPI parallel architecture provided by WRF. When running WRF-GC in the distributed-memory configuration, WRF decomposes the horizontal computational domain evenly across the available computational cores at the beginning of runtime. Each computational core has access only to its allocated subset of the full

domain as a set of atmospheric columns, plus a halo of columns around that subset domain. The halo columns are used for inter-core communication of grid-aware processes, such as horizontal transport (Skamarock et al., 2008). The internal states of GEOS-Chem for each core are managed by the state management module; they are distributed at initialization and independent from each other. The WRF-GC state management module is also critical to the development of nested-grid simulations in the future.

### 3.2.3 Compilation processes

From the user's standpoint, the installation and configuration processes for WRF-GC and WRF-Chem are similar. WRF-GC is installed by downloading the parent models, WRF and GEOS-Chem, and the WRF-GC Coupler, directly from their respective software repositories. The WRF model is installed in a top-level directory, while the WRF-GC Coupler and GEOS-Chem are installed in the `chem/` sub-directory, where the original WRF-Chem chemistry routines reside.

The standard WRF model includes built-in compile routines for coupling with chemistry, which are used by the compilation of WRF-Chem. WRF-GC uses these existing compile routines by substituting the parts pertinent to WRF-Chem with a generic chemistry interface. This substitution process is self-contained in the WRF-GC Coupler and requires no manual changes to the WRF code. As such, the installation and compilation of WRF-GC require no extra maintenance effort from the WRF developers, and WRF-GC operates as a drop-in chemical module to WRF.

When the user sets a compile option `WRF_CHEM` to 1, WRF reads a registry file (`registry.chem`) containing chemical species information and builds these species into the WRF model framework. The WRF compile script then calls the





`Makefile` in the `chem/` sub-directory to compile routines related to chemistry. We modified the `Makefile` in the `chem/` sub-directory to compile an unmodified copy of GEOS-Chem (located in `chem/gc/`) when the pre-processor switch `MODEL_WRF` is turned on. This compiles GEOS-Chem into two libraries, which can be called by WRF. The first GEOS-Chem library

(`libGeosCore.a`) contains all GEOS-Chem core routines. The second GEOS-Chem library (`libGIGC.a`) contains the GEOS-Chem column interface (`GIGC_Chunk_Mod`). The subsequent compilation process links these GEOS-Chem libraries and the WRF-to-Chemistry interface to the rest of the WRF code, creating a single WRF-GC executable (`wrf.exe`).

### 3.3 Treatment of key processes in the WRF-GC coupled model

Below we describe the operator splitting between WRF and GEOS-Chem within WRF-GC, as well as the treatments of some

290 of the key processes in the WRF-GC coupled model. The general Eulerian form of the coupled continued equation for $m$ chemical species with number density vector $\boldsymbol{n} = (n_1, ..., n_m)^T$ is

$$\frac{\partial n_i}{\partial t} = -\nabla \cdot (n_i \boldsymbol{U}) + P_i(\boldsymbol{n}) + L_i(\boldsymbol{n}) \qquad i \in [1, m] \tag{1}$$

$\boldsymbol{U}$ is the wind vector, which is provided by the WRF model in WRF-GC. The first term on the right-hand-side of Eq. 1 indicate the transport of species $i$, which include grid-scale advection, as well as sub-grid turbulent mixing and convective

transport . $P_i(\boldsymbol{n})$ and $L_i(\boldsymbol{n})$ are the local production and loss rates of species $i$, respectively (Long et al., 2015).

In the WRF-GC model, WRF simulates the meteorological variables using the dynamic equations and the initial and boundary conditions. These meteorological variables are then passed to the GEOS-Chem chemical module (Table 2) to solve the local production and loss terms of the continuity equation. Large-scale (grid-scale) advection of chemical species is grid-aware and is calculated by the WRF dynamical core. Local (sub-grid) vertical transport processes, including turbulent mixing within

300 the boundary layer and convective transport from the surface to the convective cloud top, are calculated in GEOS-Chem. Dry deposition and wet scavenging of chemical species is also calculated in GEOS-Chem. This operator-splitting arrangement is identical to that in the WRF-Chem model.

### 3.3.1 Emission of chemical species

Chemical emissions in the WRF-GC model are calculated online using the HEMCO module in GEOS-Chem (Keller et al.,

2014). For each atmospheric column, HEMCO reads in emission inventories of arbitrary spatiotemporal resolutions at runtime. Input of the emission data is parallelized through the domain decomposition process, which permits each CPU to read a subset of the data from the whole computational domain. HEMCO then regrids the emission fluxes to the user-defined WRF-GC domain and resolution at runtime. HEMCO also calculates meteorology-dependent emissions online using WRF meteorological variables. These currently include emissions of dust (Zender et al., 2003), sea salt (Gong, 2003), biogenic precursors (Guenther

et al., 2012), and soil $NO_x$ (Hudman et al., 2012). Meteorology-dependent emission of lightning $NO_x$ is not yet included in this WRF-GC version. The HEMCO module is part of the GEOS-Chem parent model and is updated together with it.





### 3.3.2 Sub-grid vertical transport of chemical species

Sub-grid vertical transport of chemical species in WRF-GC, including convective transport and boundary layer mixing, are calculated within GEOS-Chem. Convective mass fluxes are calculated in WRF using the cumulus parameterization scheme selected by the user, but the convective mass fluxes are not stored in the WRF meteorological variable array. We re-diagnosed the convective mass fluxes in the WRF-GC state conversion module using the WRF cumulus parameterization scheme selected by the user. This methodology is the same as that in the WRF-Chem model. The state conversion module currently supports the calculation of convective mass fluxes from the New Tiedtke scheme (Tiedtke, 1989; Zhang et al., 2011; Zhang and Wang, 2017) and the Zhang-McFarlane scheme (Zhang and McFarlane, 1995) in WRF (Table 2), because these two cumulus parameterization schemes are more physically-compatible with the convective transport scheme in GEOS-Chem. The diagnosed convective mass fluxes are then passed to GEOS-Chem to calculate convective transport (Allen et al., 1996; Wu et al., 2007).

Boundary-layer mixing is calculated in GEOS-Chem using a non-local scheme implemented by Lin and McElroy (2010). The boundary layer height and the vertical level and pressure information are passed from WRF to GEOS-Chem through the state conversion module. Again, this methodology is the same as that in the WRF-Chem model.

### 3.3.3 Dry deposition and wet scavenging of chemical species

Dry deposition is calculated in GEOS-Chem using a resistance-in-series scheme (Wesely, 1989; Wang et al., 1998). We mapped the land cover information in WRF to the land cover types of Olson et al. (2001) for use in GEOS-Chem.

To calculate the wet scavenging of chemical species in WRF-GC, we diagnosed the WRF-simulated precipitation variables using the microphysical schemes and cumulus parameterization schemes selected by the user (Table 2). The precipitation variables passed to GEOS-Chem include large-scale/convective precipitation production rates, large-scale/convective precipitation evaporation rates, and the downward fluxes of large-scale and convective ice/liquid precipitation. The microphysical schemes currently supported in WRF-GC include the Morrison 2-moment scheme (Morrison et al., 2009), the CAM5.1 scheme (Neale et al., 2012), the WSM6 scheme (Hong and Lim, 2006), and the Thompson scheme (Thompson et al., 2008). The cumulus parameterization schemes currently supported by the WRF-GC model include the New Tiedtke scheme (Tiedtke, 1989; Zhang et al., 2011; Zhang and Wang, 2017) and the Zhang-McFarlane scheme (Zhang and McFarlane, 1995).

## 4 Application: surface PM$_{2.5}$ over China during January 22 to 27, 2015

We simulated surface PM$_{2.5}$ concentrations over China during a severe haze event in January 2015 using both the WRF-GC model (WRF version v3.9.1.1, GEOS-Chem v12.2.1) and the GEOS-Chem Classic model (v12.2.1) in its nested-grid configuration. We compared the results from the two models against each other, as well as against surface measurements, to assess the performance of the WRF-GC model. Both WRF-GC and GEOS-Chem Classic simulations were conducted from January 18 to 27, 2015; the first four days initialized the model. Results from January 22 to 27, 2015 were analyzed.



## 4.1 Setup of the WRF-GC model and the GEOS-Chem model

Figure 2(a) shows the domain of the GEOS-Chem Classic nested-grid simulation. The GEOS-Chem Classic nested-grid simulation was driven by the GEOS-FP dataset from NASA GMAO at its native horizontal resolution of $0.25° \times 0.3125°$. The
vertical resolution of the GEOS-FP dataset was reduced from its native 72 levels to 47 levels by lumping levels in the stratosphere. The resulting 47 vertical layers extended from the surface to 0.01 hPa, with 7 levels in the bottom 1 km. Meteorological variables were updated every three hours (every hour for surface variables). Initial/boundary conditions of chemical species concentration were taken from the outputs of a global GEOS-Chem Classic simulation and updated at the boundaries of the nested-grid domain every 3 hours.

Figure 2(b) shows the domain of our WRF-GC simulation, with a horizontal resolution of 27 km $\times$ 27 km. We chose this domain and horizontal resolution for our WRF-GC simulation to be comparable to those of the GEOS-Chem Classic nested-grid simulation. There were 50 vertical levels in our WRF-GC simulation, which extended from the surface up to 10 hPa with 7 levels below 1 km. Meteorological boundary conditions were from the NCEP FNL dataset (doi:10.5065/D6M043C6) at $1° \times 1°$ resolution, interpolated to WRF vertical levels and updated every 6 hours. Initial/boundary conditions of chemical
species concentrations were identical to those used in the GEOS-Chem Classic nested-grid simulation but interpolated to WRF vertical levels and updated every 6 hours. In addition, we nudged the WRF-simulated meteorological fields with surface (every 3 hours) and upper air (every 6 hours) observations of temperature, specific humidity, and winds from the NCEP ADP Global Surface/Upper Air Observational Weather Database (doi:10.5065/39C5-Z211). Other physical options used in our WRF-GC simulation are summarized in Table 3.

Our WRF-GC and GEOS-Chem Classic simulations used the exact same chemical mechanism for gases and aerosols. Emissions in the two simulations were both calculated by the HEMCO module in GEOS-Chem and were completely identical for anthropogenic and biomass burning sources. Monthly mean anthropogenic emissions from China were from the Multiresolution Emission Inventory for China (MEIC, Li et al. (2014)) at $0.25° \times 0.25°$ horizontal resolution. The MEIC inventory was developed for the year 2015 and included emissions from power generation, industry, transportation, and residential activi-
ties. Agricultural ammonia emission was from Huang et al. (2012). Anthropogenic emissions from the rest of the Asia were from Li et al. (2017a), developed for the year 2010. Monthly mean biomass burning emissions were taken from Global Fire Emissions Database version 4 (GFED4) (Randerson et al., 2018). Emissions of biogenic species (Guenther et al., 2012), soil $NO_x$ (Hudman et al., 2012), sea salt (Gong, 2003), and dust (Zender et al., 2003) in the two simulations were calculated online by HEMCO using meteorology-sensitive parameterizations and thus slightly different. $PM_{2.5}$ mass concentrations were diag-
nosed for both simulations as the sum of masses of sulfate, nitrate, ammonium, black carbon, primary and secondary organic carbon, fine dust (100% of dust between 0 and 0.7 µm and 38% of dust between 0.7 and 1.4 µm), and accumulation-mode sea salt, taking into consideration the hygroscopic growth for each species at 35% relative humidity.





## 4.2 Validation against surface PM$_{2.5}$ measurements and comparison with the GEOS-Chem Classic simulation

Figure 2 compares the 6-day average surface PM$_{2.5}$ concentrations (January 22 00:00 UTC to January 28 00:00 UTC, 2015)

simulated by WRF-GC and GEOS-Chem Classic, respectively. Also shown are the PM$_{2.5}$ concentrations measured at 578 surface sites, managed by the Ministry of Ecology and Environment of China (www.cnemc.cn). We selected these 578 sites by (1) removing surface sites with less than 80% valid hourly measurements during our simulation period, and (2) sampling the site closest to the model grid center, if that model grid contained multiple surface sites. Both models were able to reproduce the general spatial distributions of PM$_{2.5}$ concentrations, including the higher concentrations over Eastern China relative to

Western China, as well as the hotspots over the North China Plan, Central China, and the Sichuan Basin. However, both models overestimated the PM$_{2.5}$ concentrations over Eastern China. The mean 6-day PM$_{2.5}$ concentrations averaged for the 578 sites as simulated by WRF-GC and by GEOS-Chem Classic were $117 \pm 68\ \mu g\,m^{-3}$ and $120 \pm 76\ \mu g\,m^{-3}$, respectively. In comparison, the observed mean 6-day PM$_{2.5}$ concentration averaged for the 578 sites was $98 \pm 43\ \mu g\,m^{-3}$.

Figure 3 shows the scatter plots of the simulated and observed daily average PM$_{2.5}$ concentrations over Eastern China

(eastward of 103°E, 507 sites) during January 22 to 27, 2015. We focused here on Eastern China, because the spatiotemporal variability of PM$_{2.5}$ concentrations is higher over this region. Again, both models overestimated the daily PM$_{2.5}$ concentrations over Eastern China, with WRF-GC performing better than GEOS-Chem Classic. The daily PM$_{2.5}$ concentrations simulated by WRF-GC were 29% higher than the observations (quantified by the reduced major-axis regression slope between the simulated and observed daily PM$_{2.5}$ concentration), with a correlation coefficient of r = 0.68. The daily PM$_{2.5}$ concentrations simulated

by the GEOS-Chem Classic were 55% higher than the observations, with a correlation coefficient of r = 0.72.

Our preliminary comparison above shows that the surface PM$_{2.5}$ concentrations simulated by the WRF-GC model were in better agreement with the surface observations than those simulated by the GEOS-Chem Classic nested-grid model. We found that this was partially because the WRF-GC model better represented pollution meteorology at high resolution relative to the GEOS-FP dataset. Figure 4 shows the average planetary boundary layer heights (PBLH) at 08:00 local time (00:00

UTC) and 20:00 local time (12:00 UTC) during January 22 to 27, 2015, as simulated by the GEOS-Chem Classic nested-grid model and the WRF-GC model, respectively, and compares them with the rawinsonde observations over China during this period (Guo et al., 2016). The GEOS-FP dataset generally underestimated the PBLH over the low-altitude areas of Eastern China. This led to significant overestimation of the simulated surface PM$_{2.5}$ concentrations over Eastern China, given the well-established negative correlation between PBLH and PM$_{2.5}$ concentration (Li et al., 2017b; Lou et al., 2019). In addition,

GEOS-FP severely overestimated PBLH over the mountainous areas in Southwestern China. In comparison, the WRF-GC model correctly represented the PBLH over most regions in China, which was critical to the accurate simulation of surface PM$_{2.5}$ concentrations.





## 5 Computational performance and scalability of WRF-GC

### 5.1 Computational performance of the WRF-GC model

We evaluated the computational performance of a WRF-GC simulation and compared it with that of the GEOS-Chem Classic nested-grid simulation of a similar configuration. We performed the WRF-GC and GEOS-Chem Classic simulations over the exact same domain (as shown in Figure 2(a)), with the same projection and grid sizes ($0.25° \times 0.3125°$ resolution, $225 \times 161$ grid boxes) as well as the same emissions and chemical configurations. Both simulations ran for 48 hours and used 10-minute external chemical time steps with scheduled output for every 1 hour. The WRF-GC model calculated online meteorology with

a 120-second time step, while the GEOS-Chem Classic model read in archived GEOS-FP meteorological data. In addition, WRF-GC used MPI parallelization, while GEOS-Chem used OpenMP. Both simulations executed on a single node hardware with 32 Intel Broadwell physical cores on a local Ethernet-connected file system.

Figure 5 compares the timing results for the WRF-GC and the GEOS-Chem Classic simulations. The overall wall time for the WRF-GC simulation was 5127 seconds, which was 31% of the GEOS-Chem Classic wall time (16391 seconds). We found

that the difference in computational performance was mainly due to the much faster dynamic and transport calculations in the WRF model relative to the transport calculation in the GEOS-Chem Classic. In addition, WRF-GC calculates meteorology online entirely in node memory, which eliminates the need to read archived meteorological data. In comparison, GEOS-Chem Classic reads meteorological data from disks, which poses a bottleneck. Finally, the MPI parallelization used by WRF-GC is more efficient than the OpenMP used by GEOS-Chem Classic, such that the GEOS-Chem modules actually run faster in

WRF-GC than they do in GEOS-Chem Classic. This is because OpenMP parallelization in GEOS-Chem is only at the loop level, while WRF-GC performs domain decomposition at the model level, thus parallelizing all code within the GEOS-Chem module. The WRF-GC Coupler consumed negligible wall time (39 seconds) in this test simulation.

### 5.2 Scalability of the WRF-GC model

We analyzed the scalability of the WRF-GC model using timing tests of a 48-hour simulation over East and Southeast Asia. The

425 domain size was $225 \times 161$ grid boxes (27 km $\times$ 27 km resolution). The WRF-GC simulation used the standard GEOS-Chem troposphere-stratosphere oxidant-aerosol chemical mechanism. The time steps were 120 seconds for WRF and 10 minute for GEOS-Chem chemistry (external time step), with scheduled output every hour. The WRF-GC simulation, including its input/output processes was parallelized across computational cores. The WRF-GC model was compiled using the Intel C and Fortran Compilers (v16.0.3) and the mvapich2 (v2.3) MPI library. The computing environment (Tianhe-1A) had 28 Intel

Broadwell physical cores with 125 GB of RAM per node. Input and output used a networked Lustre high-performance file system.

Figure 6 shows the scalability of our WRF-GC simulation in terms of the total WRF-GC wall time, as well as the wall times of its three components: (1) the WRF model (including input/output), (2) the GEOS-Chem model, and (3) the WRF-GC Coupler. For the domain of this test simulation, the total wall time and the WRF wall time both scale well up to 136 cores. This

is because the simulation domain becomes too fragmented above 136 cores, such that MPI communication times dominate





the run time, resulting in performance degradation. Chemical calculations in the GEOS-Chem model are perfectly scalable, consistent with previous GCHP performance analyses (Eastham et al., 2018). Figure 6 also shows that the WRF-GC Coupler scales nearly perfectly and consumes less than 1% of the total WRF-GC wall time up to 250 cores. At above 200 cores, there is a slight degradation of the scalability due to cross-core communications at the sub-domain boundaries. However, since the
440 WRF-GC Coupler is so light-weight, the impact on the total WRF-GC wall time is completely negligible.

WRF-GC also scales to massively parallel architectures and can be deployed on the cloud, because both the WRF and GEOS-Chem model are already operational on the cloud with the necessary input data readily available (Hacker et al., 2017; Zhuang et al., 2019). We conducted a preliminary test using WRF-GC on the Amazon Web Services (AWS) cloud with 32 nodes and 1152 cores. The simulation domain was over the continental United States at $5 \times 5$ km resolution with $950 \times 650$
grid boxes, with 10 second dynamical time step and 5 minute chemical time step. We found that in this massively parallel environment, the chemical wall time normalized by number of grid cells and per core was 85% of the 252-core simulation. This indicates good scalability of the chemistry component in WRF-GC. The WRF-GC Coupler took less than 0.2% of the total computational time in this simulation.

## 6 Conclusions

We developed the WRF-GC model, which is an online coupling of the WRF meteorological model and the GEOS-Chem chemical model, to simulate regional atmospheric chemistry at high resolution, with high computational efficiency, and underpinned by the latest scientific understanding of atmospheric processes. By design, the WRF-GC model is structured to work with unmodified copies of the parent models and involves no hard-wired code to either parent model. This allows the WRF-GC model to integrate future updates of both models with immediacy and ease, such that WRF-GC can stay state-of-the-science.

WRF-GC provides current users of WRF-Chem and other regional models with access to GEOS-Chem, which is state-of-the-science, well-documented, traceable, benchmarked, actively developed by a large international community, and centrally managed. GEOS-Chem users also benefit from the coupling to the open-source, community-supported WRF meteorological model. WRF-GC enables GEOS-Chem users to perform high resolution regional chemistry simulations in both forecast and hindcast mode at any location and time of interest, with high performance.

Our preliminary test shows that the WRF-GC model is able to better represent the spatiotemporal variation of surface $PM_{2.5}$ concentrations over China in winter than the GEOS-Chem Classic nested-grid model. This is because the WRF-GC model better represented the planetary boundary layer heights over the region. In addition, the WRF-GC simulation was 3 times faster than a comparable GEOS-Chem Classic simulation.

WRF-GC also scales nearly perfectly to massively parallel architectures. This enables the WRF-GC model to be used on
multiple-node systems and on supercomputing clusters, which was not possible with GEOS-Chem Classic. The GCHP model also scales to massively parallel architectures, but GCHP can only operate as a global model. Furthermore, the WRF-GC model can be deployed on the cloud, which will greatly increase WRF-GC's accessibility to new users.





The WRF-GC coupling structure, including the GEOS-Chem column interface and the state conversion module, are extensible and can be adapted to models other than WRF. This opens up possibilities of coupling GEOS-Chem to other weather and Earth System models in an online, modular manner. Using unmodified copies of parent models in coupled models reduces maintenance, avoids branching of parent model code, and enables the community to quickly and easily contribute developments in the coupled model back to the parent models.

The WRF-GC model is free and open-source to all users. The one-way coupled version of WRF-GC (v1.0) is now publicly available at wrf.geos-chem.org. A two-way coupled version with chemistry feedback to meteorology is under development and will be presented in a future paper. We envision WRF-GC to become a powerful tool for research, forecast, and regulatory applications of regional atmospheric chemistry and air quality.

*Code availability.*

WRF-GC is free and open-source and can be obtained at http://wrf.geos-chem.org. The version of WRF-GC (v1.0) described in this paper supports WRF v3.9.1.1 and GEOS-Chem v12.2.1 and is permanently archived at https://github.com/jimmielin/wrf-gc-pt1-paper-code (doi:10.5281/zenodo.3550330). The two parent models, WRF and GEOS-Chem, are also open-source and can be obtained from their developers at https://www.mmm.ucar.edu/weather-research-and-forecasting-model and http://www.geos-chem.org, respectively.





## Appendix A: Acronyms

| Acronym | Description |
| --- | --- |
| ARW | Advanced Research WRF (dynamical core) |
| CCN | Cloud condensation nuclei |
| CMAQ | Community Multiscale Air Quality Modeling System |
| CTM | Chemical transport model |
| ESMF | Earth System Modeling Framework |
| GCC | GEOS-Chem Classic |
| GCHP | GEOS-Chem High Performance |
| GCM | General circulation model |
| GDAS | Global Data Assimilation System |
| GEOS | Goddard Earth Observing System |
| GEOS-FP | GEOS Forward Processing |
| GMAO | NASA Global Modeling and Assimilation Office |
| HEMCO | Harvard-NASA Emissions Component |
| KPP | Kinetic PreProcessor |
| MAPL | Model Analysis and Prediction Layer |
| MERRA-2 | Modern-Era Retrospective analysis for Research and Applications, Version 2 |
| MMM | Mesoscale and Microscale Meteorology Laboratory, NCAR |
| MPI | Message Passing Interface |
| NCAR | National Center of Atmospheric Research |
| NCEP | National Centers for Environmental Prediction |
| NWP | Numerical weather prediction |
| PBLH | Planetary Boundary Layer Height |
| POA | Primary organic aerosol |
| SOA | Secondary organic aerosol |
| WRF | Weather Research and Forecasting Model |
| WRF-Chem | Weather Research and Forecasting model coupled with Chemistry |
| UCX | Unified Chemistry Extension |
| VBS | Volatility Basis Set |





*Author contributions.*

TMF envisioned and oversaw the project. HL designed the WRF-GC Coupler. HL, XF, and HT developed the WRF-GC
code, with assistance from YM and LJZ. XF, HL, and TMF performed the simulations and wrote the manuscript. HL performed
the scalability and analysis. RMY, MPS, EWL, JZ, DJJ, XL, SDE, and CAK assisted in the adaptation of the GEOS-Chem
model and the HEMCO module to WRF-GC. QZ provided the MEIC emissions inventory for China. XL, LZ, and LS prepared
the MEIC emissions for GEOS-Chem. JG provided the boundary layer height observations. All authors contributed to the
manuscript.

*Competing interests.*   The authors declare no competing interests.

*Acknowledgements.*   This project was supported by the National Natural Sciences Foundation of China (41975158). GEOS-FP data was
provided by the Global Modeling and Assimilation Office (GMAO) at NASA Goddard Space Flight Center. We gratefully acknowledge the
developers of WRF for making the model free and in the public domain.



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





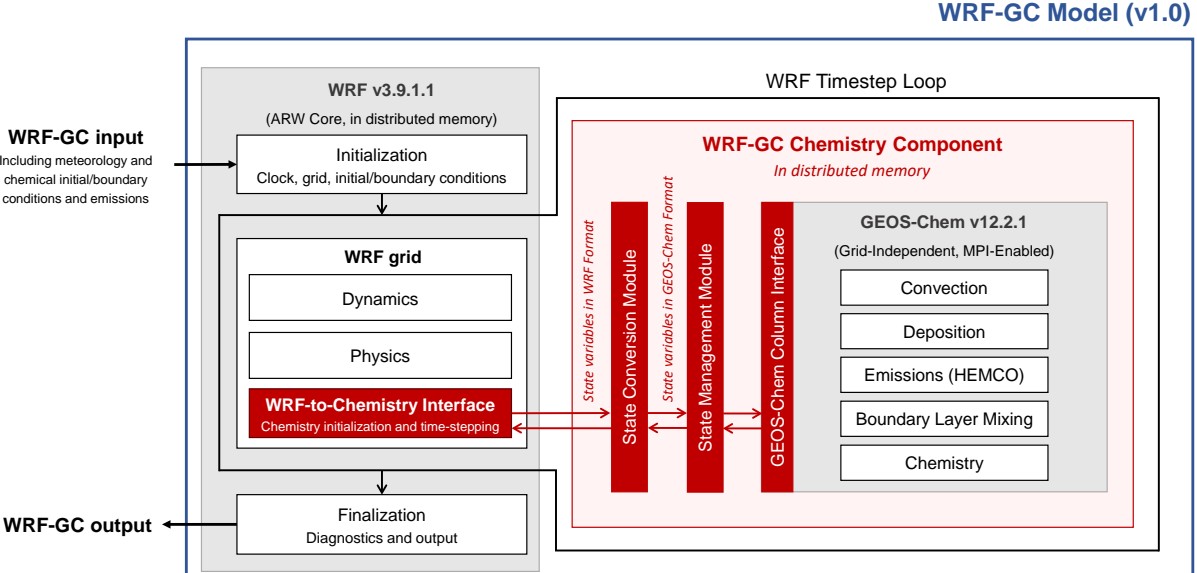

**Figure 1.** Architectural overview of the WRF-GC coupled model (v1.0). The WRF-GC Coupler (all parts shown in red) includes interfaces to the two parent models, as well as the state conversion and state management modules. The parent models (shown in grey) are standard codes downloaded from their sources, without any modifications.



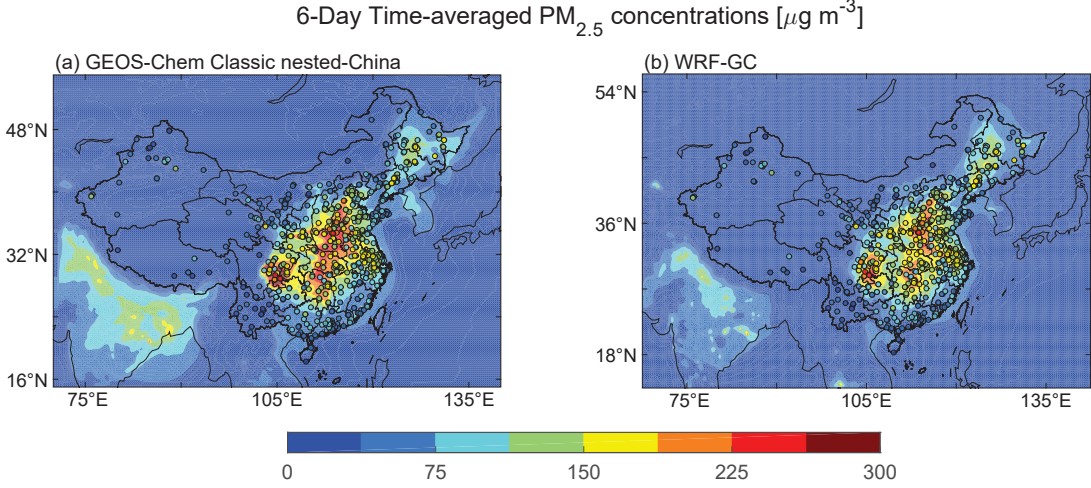

**Figure 2.** Comparison of the simulated (filled contours) 6-day average PM$_{2.5}$ concentrations during Jan 22 to 27, 2015 from (a) the GEOS-Chem Classic nested-China simulation and (b) the WRF-GC nudged simulation. Also shown are the observed 6-day average PM$_{2.5}$ concentrations during this period at 578 surface sites managed by the Ministry of Ecology and Environment of China.





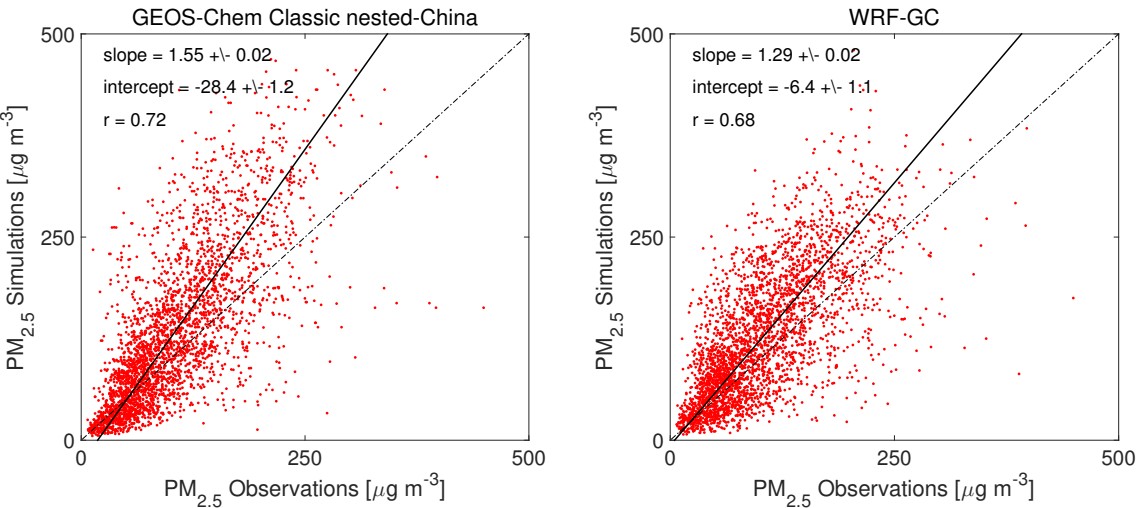

**Figure 3.** Scatter plots of observed and simulated daily mean PM$_{2.5}$ during Jan 22 to 27, 2015 at 507 surface sites over Eastern China for (a) theGEOS-Chem Classic nested-China simulation and (b) the WRF-GC nudged simulation. The solid lines indicate the reduced major axis regression lines, with slopes, intercepts, and correlation coefficients (r) shown inset. The dotted lines indicate the 1:1 lines.



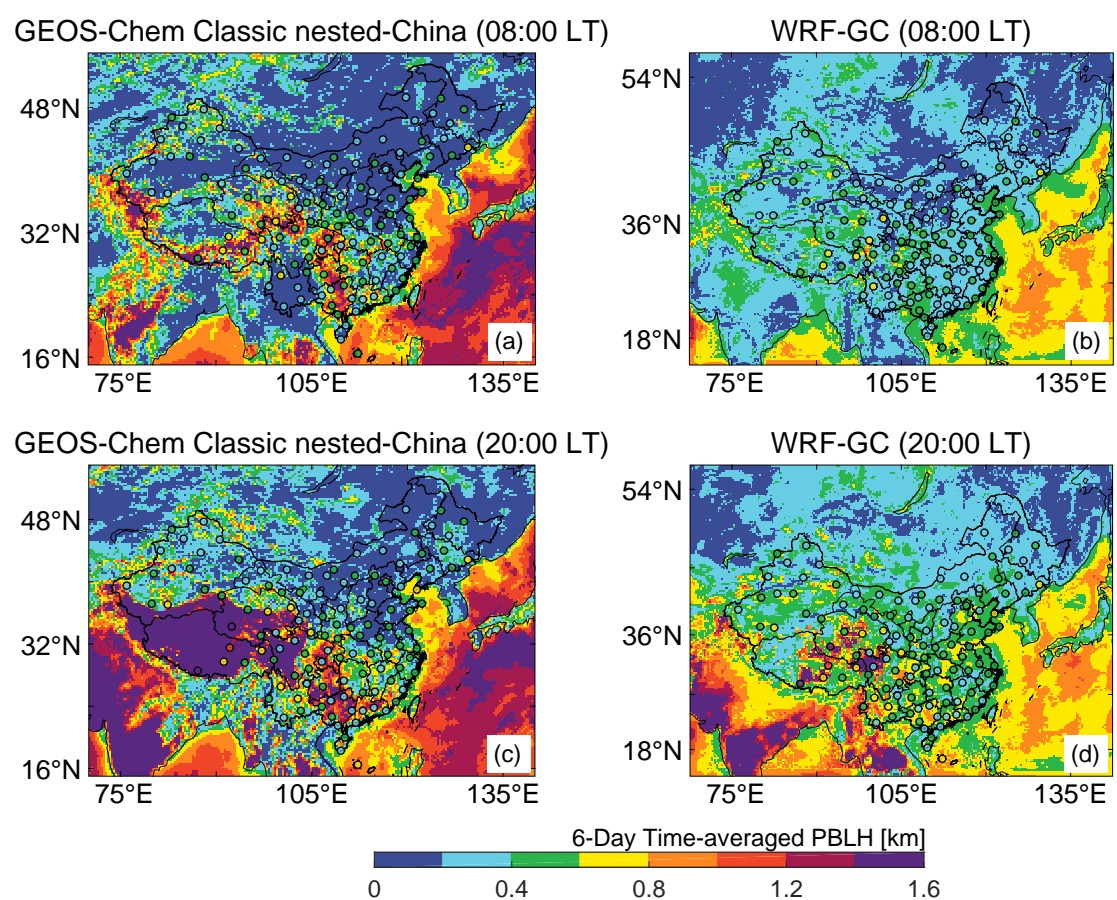

**Figure 4.** Comparison of the simulated (fill contours) and observed (fill symbols) planetary boundary layer heights (PBLH) at 08:00 local time (upper panel) and 20:00 local time (bottom panel) averaged between Jan 22 and 27, 2015. (a,c) GEOS-Chem Classic nested-China simulation (read from the GEOS-FP dataset), (b,d) WRF-GC simulation.





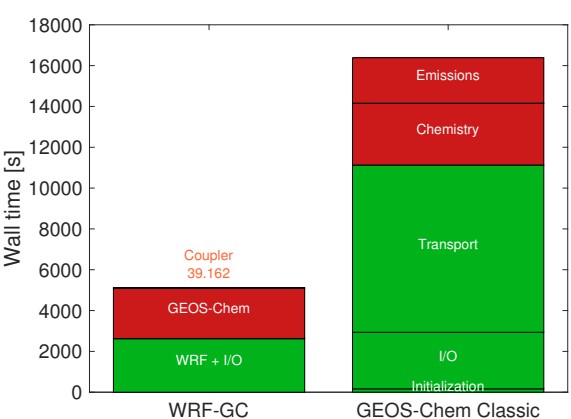

**Figure 5.** Comparison of wall time for the WRF-GC model (v1.0) and the GEOS-Chem Classic nested-grid model (version 12.2.1)





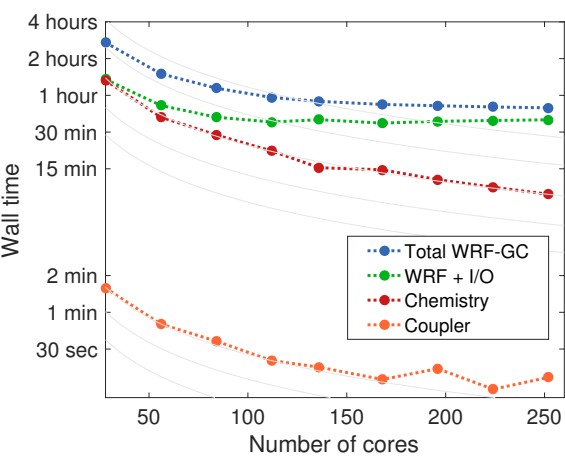

**Figure 6.** WRF-GC model scalability by processes. Gray lines indicate perfect scalability, i.e. halved computational time for each doubling of processor cores.





**Table 1.** Summary of the regional offline/online air quality models in common use

| Regional air quality model | Source of meteorological fields (A: reanalysis data; M: model) | Chemistry feedback to meteorology | Chemistry | Last 3 major updates to chemistry (date) | Licensing / charge | Number of publications during 2014-2018 from Web of Science | Reference |
|---|---|---|---|---|---|---|---|
| **Offline** | | | | | | | |
| CAMx | MM5(M), WRF(M), RAMS(M) | N | $O_3$-$NO_x$-VOC-aerosol-halogen | v6.50 (Apr 2018) v6.40 (Dec 2016) v6.30 (Apr 2016) | Open-source / free | 144 | ENVIRON, 2018 |
| CHIMERE | ECMWF(A), WRF(M) | N | $O_3$-$NO_x$-VOC-aerosol-halogen | 2017r4 (Jan 2019) 2017 (Mar 2017) 2013b (Mar 2014) | Open-source / free | 114 | Menut et al., 2013; Mailler et al., 2017; Couvidat et al., 2018 |
| CMAQ | MM5(M), WRF(M) | N | $O_3$-$NO_x$-VOC-aerosol-halogen | v5.3 (Aug 2019) v5.2.1 (Mar 2018) v5.2 (Jun 2017) | Open-source / free | 615 | Byun and Schere, 2006; Foley et al., 2010; Appel et al., 2017 |
| EMEP | MSC-W(M) | N | $O_3$-$NO_x$-VOC-aerosol | rv4.17 (Feb 2018) rv4.15 (Sep 2017) rv4.10 (Sep 2016) | Open-source / free | 176 | Simpson et al., 2012 |
| GEOS-Chem Classic (nested) | GEOS-FP(A), MERRA (A) | N | $O_3$-$NO_x$-VOC-aerosol-halogen | v12.3 (Apr 2019) v12.2 (Feb 2019) v12.1 (Nov 2018) | Open-source / free | 37 | Bey et al., 2001; |
| LOTOS-EUROS | ECMWF(A), WRF(M), RACMO(M) | N | $O_3$-$NO_x$-VOC-aerosol | v2.0 (Oct 2016) v1.10.5 | Open-source / free | 48 | Manders et al., 2017 |
| NAQPMS | MM5(M), WRF(M) | N | $O_3$-$NO_x$-VOC-aerosol | No information | Proprietary | 53 | Wang et al., 2006 |
| SILAM | HIRLAM(M), ECMWF(A) | N | $O_3$-$NO_x$-VOC-aerosol | v5.6 v5.5 v5.0 | Open-source / free | 22 | Sofiev et al., 2015 |
| TM5 | ECMWF(A), ERA-Interim(A) | N | $O_3$-$NO_x$-VOC-aerosol | TM5-MP (May 2016) v3.0 (June 2010) | Open-source / free | 36 | Huijnen et al., 2010; Krol et al., 2005; Williams et al., 2017 |





| *Online* | | | | | | | |
|---|---|---|---|---|---|---|---|
| C-IFS | ECMWF(A) | Y | $O_3$-$NO_x$-VOC-aerosol | No information | Open-source / free | 13 | Flemming et al., 2009 |
| ICON-ART | ICON(M) | Y | $O_3$-$NO_x$-VOC-aerosol | v1.0 (Dec 2014) v2.0 (Oct 2016) v2.3 (Nov 2017) | Open-source / free | 12 | Rieger et al., 2015; Weimer et al., 2017; Eckstein et al., 2018 |
| WRF-Chem | WRF(M) | Y | $O_3$-$NO_x$-VOC-aerosol-halogen | v4.1 (Apr 2019) v3.9 (May 2017) v3.8 (Apr 2016) | Open-source / free | 533 | Grell et al., 2005; Fast et al., 2006 |
| WRF-CMAQ (online) | WRF(M) | Y | $O_3$-$NO_x$-VOC-aerosol-halogen | v5.2 (Jun 2017) v5.1 (Nov 2015) v5.0 (Feb 2012) | Open-source / free | 7 | Wong et al., 2012; Yu et al, 2014 |
| WRF-GC (this work) | WRF(M) | N (v1.0) | $O_3$-$NO_x$-VOC-aerosol-halogen | Same as GEOS-Chem v12.3 (Apr 2019) v12.2 (Feb 2019) v12.1 (Nov 2018) | Open-source / free | - | This work |





**Table 2.** Meteorological variables required to drive GEOS-Chem that are passed or calculated from the WRF model by the WRF-GC Coupler

| No. | Variable(s) in GEOS-Chem [unit] | Description | Usage in GEOS-Chem | Passed or calculated from which variable(s) in WRF [unit] |
|---|---|---|---|---|
| *Treatment in Coupler: passed from WRF without change* | | | | |
| 1 | ALBD [unitless] | Visible surface albedo | Dry deposition | ALBEDO [unitless] |
| 2 | CLDF [unitless] | 3-D cloud fraction | Photolysis; chemistry | CLDFRA [unitless] |
| 3 | CLDFRC [unitless] | Column cloud fraction | Photolysis | CLDT [unitless] |
| 4 | EFLUX [W m$^{-2}$] | Latent heat flux | Diagnostics | LH [W m$^{-2}$] |
| 5 | FRSEAICE [unitless] | Fraction of sea ice | Hg simulation | FRSEAICE [unitless] |
| 6 | GWETROOT [unitless] | Root soil wetness | Diagnostics | SM100200 [m$^3$ m$^{-3}$] |
| 7 | GWETTOP [unitless] | Top soil moisture | CH$_4$ simulation; dust mobilization | SM000010 [m$^3$ m$^{-3}$] |
| 8 | HFLUX [W m$^{-2}$] | Sensible heat flux | Dry deposition | HFX [W m$^{-2}$] |
| 9 | LAI [m$^2$ m$^{-2}$] | Leaf area index | Diagnostics | LAI [m$^2$ m$^{-2}$] |
| 10 | PBLH [m] | Planetary boundary layer height | PBL mixing | PBLH [m] |
| 11 | PFILSAN [kg m$^{-2}$ s$^{-1}$] | Downward flux of large-scale + anvil ice precipitation | Wet scavenging; | PRECR [kg m$^{-2}$ s$^{-1}$] |
| 12 | QI [kg kg$^{-1}$] | Cloud ice water mixing ratio | Chemistry; aerosol microphysics | QI [kg kg$^{-1}$] |
| 13 | QL [kg kg$^{-1}$] | Cloud liquid water mixing ratio | Chemistry; aerosol microphysics | QC [kg kg$^{-1}$] |
| 14 | SNODP [m] | Snow deposition | Diagnostics | SNOWH [m] |
| 15 | SNOMAS [kg m$^{-2}$] | Snow mass | Dust mobilization; Hg simulation; dry deposition; | ACSNOW [kg m$^{-2}$] |
| 16 | SWGDN [W m$^{-2}$] | Surface incident radiation | Soil NO$_x$ emissions; Hg simulation; dry deposition | SWDOWN [W m$^{-2}$] |
| 17 | TS [K] | Surface temperature | Many locations | T2 [K] |
| 18 | TSKIN [K] | Surface skin temperature | CH$_4$ simulation; Hg simulation; sea salt emissions | TSK [K] |
| 19 | U [m s$^{-1}$] | East-west component of wind | Advection | U [m s$^{-1}$] |
| 20 | USTAR [m s$^{-1}$] | Friction velocity | Dry deposition | UST [m s$^{-1}$] |
| 21 | U10M [m s$^{-1}$] | East-west wind at 10m height | Dry deposition; dust mobilization; Hg simulation; sea salt emissions | U10 [m s$^{-1}$] |
| 22 | V [m s$^{-1}$] | North-south component of wind | Advection | V [m s$^{-1}$] |
| 23 | V10M [m s$^{-1}$] | North-south wind at 10m height | Dry deposition; dust mobilization; Hg simulation; sea salt emissions | V10 [m s$^{-1}$] |
| 24 | Z0 [m] | Surface roughness height | Dry deposition | ZNT [m] |





| | | | | |
|---|---|---|---|---|
| *Treatment in Coupler: converted into GEOS-Chem units or diagnosed from WRF variables* | | | | |
| 25 | AREA_M2 [m$^{-2}$] | Grid box surface area | Many locations | DX/DY (X/Y horizontal resolution) [m]; MSFTX/MSFTY (Map scale factor on mass grid, x/y direction) [unitless] |
| 26 | CMFMC [kg m$^{-2}$ s$^{-1}$] | Cloud mass flux | Convective transport | MFUP_CUP [kg m$^{-2}$ s$^{-1}$]; CMFMCDZM [kg m$^{-2}$ s$^{-1}$]; CMFMC [kg m$^{-2}$ s$^{-1}$] |
| 27 | DQRCU [kg kg$^{-1}$ s$^{-1}$] | Convective precipitation production rate | Wet scavenging (in convective updraft) | DQRCU [kg kg$^{-1}$ s$^{-1}$] |
| 28 | DQRLSAN [kg kg$^{-1}$ s$^{-1}$] | Large-scale precipitation production rate | Wet scavenging | RAINPROD [kg kg$^{-1}$ s$^{-1}$]; PRAIN3D [kg kg$^{-1}$ s$^{-1}$]; |
| 29 | DTRAIN [kg m$^{-2}$ s$^{-1}$] | Detrainment flux | Convective transport | DU3D [s$^{-1}$]; DTRAIN [kg m$^{-2}$ s$^{-1}$] |
| 30 | FRLAKE [unitless]; FRLAND [unitless]; FRLANDIC [unitless]; FROCEAN [unitless]; FRSNO [unitless]; | Fraction of land/ocean/surface snow/lake/land ice | Chemistry; Hg simulation; CH$_4$ simulation; PBL mixing; emissions; diagnostics | LU_MASK (0-land, 1-water) [unitless]; LAKEMASK [unitess]; SNOWH [m] |
| 31 | LANDTYPEFRAC [unitless] | Olson fraction per land type | Dry deposition | LU_INDEX (land use category) [unitless] |
| 32 | LWI [unitless] | Land-water-ice indices | Many locations | LU_MASK [unitless] |
| 33 | OMEGA [Pa s$^{-1}$] | Updraft velocity | Diagnostics | W [m s$^{-1}$] |
| 34 | OPTD [unitless] | Visible cloud optical depth | Photolysis; chemistry | TAUCLDI [unitless]; TAUCLDC [unitless] |
| 35 | PARDF [W m$^{-2}$] | Diffuse photosynthetically active radiation | Biogenic emissions | SWVISDIF (Diffuse photosynthetically active radiation) [W m$^{-2}$]; P (perturbation pressure) [Pa]; PB (base state pressure) [Pa]; COSZEN (cosine of solar zenith angle) [unitless]; SWDOWN [W m$^{-2}$] |



| 36 | PARDR [W m$^{-2}$] | Direct photosynthetically active radiation | Biogenic emissions | SWVISDIR (Direct photosynthetically active radiation) [W m$^{-2}$]; SWDOWN [W m$^{-2}$]; P [Pa]: PB [Pa]; COSZEN [unitless] |
|---|---|---|---|---|
| 37 | PEDGE [hPa] | Wet air pressure at level edges | Many locations | PSFC [Pa]; P_TOP [Pa]; C3F [unitless]; C4F [unitless] |
| 38 | PFICU [kg m$^{-2}$ s$^{-1}$] | Downward flux of convective ice precipitation | Wet scavenging (in convective updraft) | PMFLXSNOW [kg m$^{-2}$ s$^{-1}$] |
| 39 | PFLCU [kg m$^{-2}$ s$^{-1}$] | Downward flux of convective liquid precipitation | Wet scavenging (in convective updraft) | PMFLXRAIN [kg m$^{-2}$ s$^{-1}$] |
| 40 | PFLLSAN [kg m$^{-2}$ s$^{-1}$] | Downward flux of large-scale + anvil liquid precipitation | Wet scavenging | PRECI [kg m$^{-2}$ s$^{-1}$]; PRECS [kg m$^{-2}$ s$^{-1}$] |
| 41 | PHIS [m$^2$ s$^{-2}$] | Surface geopotential height | Diagnostics | PHB (base state geopotential) [m$^2$ s$^{-2}$]; PH (perturbation geopotential) [m$^2$ s$^{-2}$] |
| 42 | PRECANV [kg m$^{-2}$ s$^{-1}$] | Anvil precipitation | Diagnostics | SNOWNCV/GRAUPELNCV/HAILNCV (time-step non-convective snow and ice/graupel/hail) [mm] |
| 43 | PRECCON [kg m$^{-2}$ s$^{-1}$] | Surface convective precipitation | Soil NO$_x$ emissions; wet scavenging | PRATEC [mm s$^{-1}$] |
| 44 | PRECLSC [kg m$^{-2}$ s$^{-1}$] | Non-anvil large-scale precipitation | Diagnostics | RAINNCV (time-step non-convective rain) [mm] |
| 45 | PRECTOT [kg m$^{-2}$ s$^{-1}$] | Surface total precipitation | Soil NO$_x$ emissions; wet scavenging | RAINNCV/SNOWNCV/GRAUPELNCV/HAILNCV [mm]; PRATEC [mm s$^{-1}$] |
| 46 | PS1DRY [hPa] | Dry surface pressure at dt start | Advection; many other locations | PSFC [Pa] |
| 47 | REEVAPCN [kg kg$^{-1}$ s$^{-1}$] | Evaporation of convective precipitation | Wet scavenging (in convective updraft) | REEVAPCN [kg kg$^{-1}$ s$^{-1}$] |





| 48 | REEVAPLS [kg kg⁻¹ s⁻¹] | Evaporation of large-scale + anvil precipitation | Wet scavenging | EVAPPROD [kg kg⁻¹ s⁻¹]; NEVAPR3D [kg kg⁻¹ s⁻¹] |
|---|---|---|---|---|
| 49 | RH [%] | Relative humidity | Chemistry; wet scavenging; Aerosol thermal equilibrium; Aerosol microphysics | T (perturbation potential temperature) [K]; QV (water vapor mixing ratio) [kg kg⁻¹]; P [Pa]; PB [Pa] |
| 50 | SPHU [g kg⁻¹] | Specific humidity | Chemistry; wet scavenging; PBL mixing | QV [kg kg⁻¹] |
| 51 | T [K] | Temperature | Many locations | T [K]; P [Pa]; PB [Pa] |
| 52 | TAUCLI [unitless] | Optical depth of ice clouds | Diagnostics | TAUCLDI (Optical depth of ice clouds) [unitless]; T [K]; P [Pa]; PB [Pa]; QI [kg kg⁻¹] |
| 53 | TAUCLW [unitless] | Optical depth of water clouds | Diagnostics | TAUCLDC (Optical depth of water clouds) [unitless]; T [K]; P [Pa]; PB [Pa]; QC [kg kg⁻¹]; QNDROP (droplet number mixing ratio) [# kg⁻¹] |
| 54 | TO3 [DU] | Total overhead O₃ column | Photolysis | O3 [ppmv] |
| 55 | TROPP [hPa] | Tropopause pressure | Tropopause height diagnosis | TROPO_P [Pa] |
| 56 | XLAI [unitless] | MODIS LAI per land type | Dry deposition | LAI [unitless]; LU_INDEX [unitless] |





**Table 3.** WRF-GC physics configuration.

|  | Physical Options |
| --- | --- |
| Microphysics | Morrison 2-moment (Morrison et al., 2009) |
| Longwave radiation | RRTMG (Iacono et al., 2008) |
| Shortwave radiation | RRTMG (Iacono et al., 2008) |
| Surface layer | MM5 Monin-Obukhov (Jimenez et al., 2012) |
| Land surface | Noah (Chen and Dudhia, 2001a, b) |
| Planetary boundary layer | MYNN2 (Nakanishi and Niino, 2006) |
| Cumulus | New Tiedtke (Tiedtke, 1989; Zhang et al., 2011; Zhang and Wang, 2017) |