# Peer review of "WRF-GC (v1.0): online coupling of WRF (v3.9.1.1) and GEOS-Chem (v12.2.1) for regional atmospheric chemistry modeling, Part 1: description of the one-way model"

_Geoscientific Model Development, 2019_

## Short Comment (SC1) · 23 Jan 2020

Dear authors,

in my role as Executive editor of GMD, I would like to bring to your attention our Editorial version 1.2:

https://www.geosci-model-dev.net/12/2215/2019/

This highlights some requirements of papers published in GMD, which is also available on the GMD website in the 'Manuscript Types' section:

http://www.geoscientific-model-development.net/submission/manuscript_types.html

[Figure]

In particular, please note that for your paper, the following requirement has not been met in the Discussions paper:

- "The main paper must give the model name and version number (or other unique identifier) in the title."

Please add a version number WRF and GEOS-CHEM in the title upon your revised submission to GMD (as named in the code availability section).

Yours,

Astrid Kerkweg
* * *

---

## Referee Comment (RC1) · Anonymous Referee #1 · 11 Mar 2020

The manuscript 'WRF-GC: online coupling of WRF and GEOS-Chem for regional atmospheric chemistry modeling, Part 1: description of the one-way model (v1.0)' written by Haipeng Liu and group team presented the development of regional chemical transport model coupled with global chemistry model of GEOS-Chem. The authors described the method of coupling of GEOS-Chem to WRF, and further conducted the test case over China, and compared model performances and computational time. Although I would like to consider the publication of this attractive research to develop the sophisticated regional chemical transport models, I hope the manuscript is fully revised to address following comments.

Major comments:

Introduction:

As the introduction in this study, the authors well summarized CTMs in terms of off-line and on-line models, and also picked up some models such as CMAQ and WRF-Chem to discuss. Because the application of WRF-GC to show the performance improvements was just conducted in comparison with GEOS-Chem Classic, and without the direct comparison with other CTMs summarized in Table 1 (especially WRF-Chem as well stated in the manuscript), it is not appropriate to emphasize the superiority of WRF-GC model. I would like to suggest to focus only on GEOS-Chem Classic model in introduction section. For example, it seems to be one of the significant point in terms of the necessity of fine-scale simulation beyond the Classic nested version. In addition, I am not GEOS-Chem user, but the statement posed me some biases in GEOS-Chem. Especially, some criticism on WRF-Chem model can be avoided as much as possible without appropriate evidences.

Application:

Related above mentioned comment, without the evaluation to other models, this study should keep focus on the comparison between GEOS-Chem Classic and developed WRF-GC in terms of the model superiority. In addition, the application is just one case over China and only six days, it should be noted and soften the conclusion derived in this study. In this application, I have following two comments.

1. Sections 4.1 and 4.2:

GEOS-Chem Classic nested version includes 47 layers with 7 levels in the bottom of 1 km, whereas WRF-GC had 50 layers with 7 levels in the bottom of 1 km. Are these 7 levels (or surface level) identical? Even a few meters differences could cause the modeling performance difference, and I am suspicious the conclusion attributed to the behavior of planetary boundary layer. If the configuration of boundary layer is identi-

cal, the height of planetary boundary layer could be one reason as model difference; however, what are other basic meteorological parameters such as wind speed and its direction, temperature, relative humidity, and precipitation (related to wet scavenging)? More investigations on meteorological fields are required in this part.

2. Section 5.1:

Again, what is the configuration of vertical layer in this WRF-GC? Without the identical setting of it, the comparison in computational performance is not appropriate. Under the same configuration of both models, the performance differences can be attributed to the importance of online and offline simulation. In addition to the behavior of meteorological fields, how about the importance of online and offline model?

Minor comments:

P6, L175-L179: Does WRF-GC cover both yield model and VBS model for SOA? It is not explicitly stated which is available.

P6, L179-180: What is size bins for sulfate, nitrate, ammonium, black carbon, and POA and SOA? Only dust and sea salt is described here.

P7, L203: How can we prepare the initial and boundary conditions for chemical variables? It is not well stated in the current document.

P7, L206: What is the "choice of chemical species"? In P6, L163-164, it is only stated "241 chemical species and 981 reactions". Do we need to set chemical species in model simulation?

P7, L206: What is the "chemical mechanisms"? In P6, L160-161, we can see "the standard chemical mechanism in GEOS-Chem", but what is other options in GEOS-Chem?

P7, L206: What is the meaning of diagnostics? This wording is ambiguous.

P8, L230: This means that WRF-GC only supports the newly established WRF vertical

grids, and does not have the choice to old terrain-following grids?

P10, L310-311: So what is the available lightning NOx emissions in current WRF-GC? It is not available, or using climatorogical value?

P11, L317-319 and P11, L328-335: From these paragraphs, I understand that the WRF-GC simulation have the limitations in the selection of various WRF options. This information is necessary to understand the limitation of applicability of WRF-GC model. Therefore, I would like to strongly suggest to put table for the summary of available options in WRF for WRF-GC.

P11, L327: For WRF model itself, the land cover can be obtained by both USGS and MODIS, but WRF-GC can be available both datasets? Please explicitly stated for the available land cover information.

P13, L391 and P25, L460: I understand that this is one test case, but the comparison should not be "preliminary".

L13, L391-L402 and Figure 4: This paragraph needs amendments in its expression. There is two expressions of "GEOS-FP dataset" and "GEOS-Chem Classic nested-China". I understand that GEOS-Chem model uses GEOS-FP; however, it seems to be better to unify its expression for readability.

P14, L424-431: I guess that this WRF-GC model for the discussion of scalability is same model in Section 5.1. If so, this paragraph contains some redundant information. Please clarify and repetition should be avoided.

Figure 1: In left corner of "WRF-GC input", there is the statement on "emissions". What is the difference of this emissions and HEMCO?

Figures 2 and 4: The map line is thinner, so please enhance the map line for readability.

Figure 6: Gray lines is hard to see. Please change the thickness of lines, or change into black color for readability.

Technical comments:

L2 in the caption of Figure 3: Need space ("theGEOS-Chem").

---

## Referee Comment (RC2) · Anonymous Referee #2 · 13 Mar 2020

This paper describes how GEOS-Chem has been implemented in the WRF model. The topic and the description of its implementation is appropriate for the GMD audience. There are certain details that need elaboration for future users of the code and I have made several suggestions. In addition, some the statements are clearly biased that overstate the capabilities of the model without actual proof. At times, the text sounds more salesmanship than scientific. Both of these concerns should be relatively easy to address.

Major Comments:

1) Boundary layer mixing is handled separately by GEOS-Chem. I am a concerned
about this approach which is glossed over. Most, but not, all of the meteorology from WRF is used and that point is passed over in other parts of the text. WRF-Chem uses parameters from the WRF boundary layer parameterizations so that the vertical mixing is treated in as similar as possible. In this way, mixing of chemical species occurs within the same boundary layer depth as the meteorology. It is not clear whether vertical mixing in GEOS-Chem and WRF are consistent. If not, it may be possible for GEOS-Chem to have a deeper or shallower boundary layer than in WRF. If deeper, than more chemistry variables will be transported by free tropospheric air and boundary layer air. The authors should delve into this in more detail to let users know what the approach for vertical mixing actually is and the potential consequences. This applies to the results shown in Figure 4.

2) At several places in the manuscript, the authors point out the reasons for the advantages of coupling GEOS-Chem within WRF. For the GEOS-Chem community the advantages are obvious. But it is not as clear what the WRF-Chem community gains. The WRF-Chem community would have another chemistry option, but this paper does not explore the types of chemical processes that might be missing (perhaps halogen chemistry) already in the model. The current aerosol treatment is rather simple (is it even "state-of the science"?) and similar in many respects to existing simple aerosol options in WRF-Chem. Users choose particular treatments in a community model, such as WRF, for various reasons such as overall performance (i.e. how well the model represents reality), physics complexity, and computational considerations. The authors have failed to articulate what the advantages are to the WRF community. The future developments beyond v1.0 may change this picture and make that argument more apparent. But if the authors want to frame their arguments in this paper, then a few more concrete points of the advantages to the WRF community are needed.

Specific Comments:

Lines 7-8: "is designed to by easy to use, . . . extendable, and easy to update" are relative terms and depends on an individual's point of view. This is something I'm

struggling with here. "is designed" is probably the key phrase. It is hard for a reviewer to verify the last part of the sentence without using the code itself. WRF itself is a rather complex model, although users with "sufficient" expertise in atmospheric models and computational hardware can learn how to run the model in a short period (i.e. days) of time. Those without "sufficient" expertise, might not describe it as easy to use. Nor am I sure what "extendable" means here. All computational models by their very nature are extendable by modifying code.

Lines 15-17. I have some concerns regarding the statement on PBL heights, in which I will comment on later in the appropriate section.

Lines 18-19: The sentence "Both parent models . . ." is redundant with an earlier statement. It probably can be deleted and the thought merged with the earlier statement.

Lines 27-28: "regional" is used twice in this sentence and is redundant and is self-evident.

Lines 33-34: "to better serve the public, inform policy makers, and advance science" is a laudable goal. However, I think these types of models are used primarily to "advance science". WRF has been used to provide short-range operational forecasts (e.g. HRRR model), but NOAA is phasing out the use of WRF. WRF-CMAQ has been serving EPA regulatory interest in addition to advancing science, but I am not sure GEOS-Chem serves that purpose.

Lines 73-86: The authors state that online models are more difficult to keep up to date than offline models. I disagree. There are many factors that contribute to how quickly and the frequency of updates made in models (some described by the authors) that have nothing to do with whether the model is online or offline. The authors first mention resources used to provide benchmarking, validating, and documentation, which is probably a good thing for both offline and online models. This process does take longer for more complex models, but offline models can be complex too. It is also not a good idea to translate research findings into publicly available without due diligence. Sometimes new scientific findings are proven to be incorrect and/or the findings are based on a limited case study; therefore, it is not wise or appropriate for use by a broad community. The authors second point on expertise residing in different communities. I agree this is an issue, but this is largely governed by how the organization wishes to develop the code, and not whether the model is offline or online. In the case of WRF, it has always been a community model so that contributions originate from various organizations and universities. Nor has NCAR insisted that all physics schemes be compatible with each other, and sometimes there are good reasons why some physics schemes should not be made compatible. Had NCAR decided to be the sole developer, the code would no doubt be more streamlined but would probably lack scientific options many users want. It seems strange to me that the authors are arguing that bring communities together introduces problems, when the purpose of WRF-GC is yet another iteration of bringing very different communities together. The authors correctly point out that not everything in WRF-Chem (and WRF for that matter) is compatible, but it is not clear whether the same holds for the various treatments in GEOS-Chem.

Lines 98-98. This sentence is kind of a put-down to WRF (in the context of this paragraph) and other models listed in Table 1. As with another statement in the introduction, this depends on one's point-of-view and difficult to prove. What is "state-of-the-science" anyway? This overused phrase is almost meaningless at this point. For examples, some chemistry and aerosol disciplines are changing very rapidly (weekly) and it is very difficult to argue which model has the most "up-to-date" science. I suggest the authors rephrase this sentence to be a bit more fair and unbiased.

Line 100: The authors state that that GEOS-Chem is driven by the meteorological fields simulated by WRF. But after reading material later in the paper, this may not be entirely true. It seems that GEOS-Chem still uses its own turbulent vertical mixing which could be different from WRF. I have some other comments on this point later in the paper.

Lines 168-169: Based on this sentence, it sounds like the bulk treatment is similar to GOCART. But the last sentence in the paragraph implies a modal treatment for dust

and sea-salt. So is dust and sea-salt in bulk bins or actually prescribed using a size distribution? Just need to be consistent in the text here.

Lines 169-172: These two sentences might be better at the end of the paragraph. It would be better to have all the discussion on the present aerosol model together, rather than broken up with what is not in the code in the middle.

Lines 181-185: I think some discussion is needed regarding higher-resolution emission datasets. GEOS-Chem has been used traditionally at global scales and there are several global emission datasets available. But if the point of WRF-GC is to run at higher resolution, this would be defeated in part by an emission inventory that is much coarser than what could be simulated by the model. What is the strategy for that? Will other inventories used by EPA and the WRF-Chem community be used? Or are users expected to generate emissions on their own? Also, emissions are discussed in more detail later in Section 3.3.1. So I am wondering if it is even necessary to talk about emissions here. This material can be merged into Section 3.3.1

Line 198: I am not sure what "greater modularity" means here. WRF-Chem is structured in a way to have modularity for important chemistry and aerosol processes. For example, there are several option for online emissions, deposition, etc. The modularity also permits users to add their own treatments if they wish. In some cases, there are treatments that can be used for multiple chemistry or aerosol options. For example, wet scavenging can be handled similarly for MADE-SORGAM (modal aerosols) and MOSAIC (sectional aerosols). But I doubt if the modules in GEOS-Chem can be used or accessed by other treatments in the code, given the structure on how it was implemented. So, if the authors want to use the phrase "greater modularity" some description is need to know exactly what this means.

Line 206: What is the "WRF-to-chemistry interface". Is GEOS-Chem handled in the registry.chem file, much like the other chemistry options? Given Figure 1, I think this will be discussed later and it looks like the material is in Section 3.2.2. I just found this

a bit vague at this point.

Line 213: See the first major comment above.

Line 220: In Section 3.2, the text implies that GEOS-chem is compiled and run on top of the host WRF model. The text implies but does not explicitly state that the addition of GEOS-Chem has not impaired the use of WRF-Chem. One of the rules in the WRF community is that any new additions must be shown to not "break" other parts of the code. So have the authors performed tests with the new code to ensure that other chemistry options produce the same results went the code is compiled with GEOS-chem?

Lines 312-335: As the authors imply, subgrid and removal processes, will depend on both "resolved" and "unresolved" clouds. In contrast with global models, simulations at fine enough resolutions may be best run with any "unresolved" cloud parameterizations. Ideally, a cumulus parameterization would have a progressively smaller and smaller impact at higher and higher resolutions. The subgrid and removal processes depend on the current behavior of what is in WRF. It would be useful to provide some discussion for users regarding the implications of these assumptions. Since this paper describes a new modeling framework, it would have been interesting to see some simulations at coarse and fine resolutions to demonstrate the differences on the vertical transport and wet scavenging. WRF-GC users will be able to run at smaller grid spacings, which is a good thing, but will they understand the subtleties of these assumptions that are not normally encountered at global scales?

Lines 323-325: Same comment as before regarding vertical mixing. This is a different treatment than in WRF. So there could be mis-matches in how boundary layer mixing, and it impact the vertical extent of boundary layer mixing, between WRF (which is used to transport chemical species) and GEOS-Chem. Lines 350-359: The authors should include the meteorology simulation time step and chemical time step. What is not mentioned in the discussion of the model's implementation, but alluded to in the

conclusion, is that a different chemical time step can be used (which is similar to WRF-Chem). In this way, transport of chemical species are done at the meteorological time step. Chemistry is usually simulated at a coarser time step to save computer time, but some care is needed since chemical time steps that are too large could introduce uncertainties in the predictions – especially at higher spatial resolutions. Since the paper is about WRF-GC and designed to work at higher spatial resolution, some discussion is needed in the best practice for the two types of time steps. Ideally, they should be the same for consistency.

Line 353: The authors use FNL for the meteorological initial and boundary conditions. To be more consistent with the nested GEOS-Chem, why not use the meteorological conditions from that model?

Lines 356-359: It sounds like the authors are using the obs-nudging capability in WRF. Would be useful to cite a reference on that. What is not clear in the previous paragraph, for readers not familiar with GEOS-Chem, that the meteorological fields are prescribed analyses. Thus nudging in WRF would be appropriate to make the meteorology in the two simulations more compatible. The authors should be more specific on these points.

Line 363-364: Do the anthropogenic emissions vary diurnally? This seems to be an important point in simulating diurnal and peak concentrations. Here the authors only show a 6-day average. Perhaps this is okay for demonstrate WRF-GC compared to GEOS-Chem, and this distinction should be made.

Lines 394: Are these boundary layer heights the same as predicted by WRF? Or different calculations in the GEOS-Chem modules?

Lines 397-402: The 6-day averaging may be hiding some day-to-day variations when trying to assess the cause of the positive PM2.5 bias. While it is very likely that the boundary layer height issue is contributing to that, there are likely other differences in the meteorology that could also be attributing factors. The authors should at least acknowledge that if they do not wish to pursue a more detailed analysis of the simulations. I assume the same boundary layer scheme is being used in both models, but the differences arise in how that scheme is driven. I presume in GEOS-Chem the meteorology is from the large-scale analyses. Whereas, in WRF the prognostic land-surface model will be controlling the evolution of surface temperature and fluxes that will drive a boundary layer parameterization. So, some additional discussion as to why these differences in the boundary layer height would be useful.

Line 417: Eliminating the need to read archived meteorology becomes even more important at high resolution offline approaches as more and more time is required for I/O.

Line 418: I do not fully understand what reading from disks means. Do the authors mean the analyses are on some long-term storage device (tapes?) in which reading is slower than conventional hard-drives? Would a fairer timing test include having meteorological fields in a place with faster access?

Line 432: Figure 6 is difficult to interpret because of the scale on the y-axis. Ideally if one doubled the number of cores, one would want the wall clock time to be reduced by a factor of 2. It is hard to see this with the current scaling. To me it looks like the total performance does not improve much greater than 100 cores. In addition to "fragmentation" the authors mention, the most CPU time in WRF is due to the advection of species. The more chemical species there are the slower the code is. The authors mention 241 species on line 163 (presumably trace gas), but there would be aerosol species on top of that. This is more than other options in WRF-Chem, but less than others. In the conclusion the paper states that the WRF-chem community would benefit from GEOS-Chem, but it is not clear what the computational cost would be compared to other approaches. A large fraction of the users in the WRF community chose "simple" chemistry and aerosol schemes to save computational cost. While the authors may not wish to perform identical simulations with different chemical options to benchmark GEOS-Chem with others, the least they could do is compare the number of species to establish some sort of computational level of complexity.

Line 440: Change "light-weight" to "efficient".

Line 449: The conclusion would benefit (perhaps at the end) about future directions. They are alluded to elsewhere in the text, but it would be good to briefly summarize them here. That would include things like more complex aerosol treatments and fully-taking advantage of on-line coupling to include feedback effects (both of which are already in WRF-Chem). The conclusion would also be strengthened to foreshadow what would be discussed in part 2.

Lines 460-463. As I mentioned earlier, the evaluation is rather simple. This paragraph should be rephrased to indicate this is only one case study (and thus not comprehensive) and other meteorological differences may be contributing to the PM2.5 predictions.

Line 464: The first sentence is an overstatement and deceiving if a reader only looks at the conclusions. Yes, the chemistry part itself scales well but not the entire code.

---

## Author Comment (AC1) · 24 Apr 2020

**Response from authors**

Re: Review of WRF-GC: online coupling of WRF and GEOS-Chem for regional atmospheric chemistry

modeling, Part 1: description of the one-way model (v1.0)

April 24, 2020

We thank the two Reviewers and the Executive Editor for their helpful comments. In response, we have carefully

revised the manuscript to (1) better focus the scope of this work, (2) supplement technical details of the WRF-GC

coupled model, (3) expand the comparison of the $PM_{2.5}$ concentrations simulated by the WRF-GC and GEOS-Chem

models, and (4) improve logic and clarity. We also revised the title of the paper to indicate the version numbers of

the two parent models.

We respond to each specific comment in detail below. The referee comments are shown in red italics. Our replies

are shown in black and modified text is shown in blue. The annotated page and line numbers refer to the revised

copy of the manuscript.

**1 Reviewer #1**

*The manuscript 'WRF-GC: online coupling of WRF and GEOS-Chem for regional atmospheric chemistry mod-*

*eling, Part 1: description of the one-way model (v1.0)' written by Haipeng Liu and group team presented the*

*development of regional chemical transport model coupled with global chemistry model of GEOS-Chem. The au-*

*thors described the method of coupling of GEOS-Chem to WRF, and further conducted the test case over China,*

*and compared model performances and computational time. Although I would like to consider the publication of*

*this attractive research to develop the sophisticated regional chemical transport models, I hope the manuscript is fully revised to address following comments.*

*Major comments:*

*R1.1 Introduction: As the introduction in this study, the authors well summarized CTMs in terms of off-line and on-line models, and also picked up some models such as CMAQ and WRF-Chem to discuss. Because the application of WRF-GC to show the performance improvements was just conducted in comparison with GEOS-Chem Classic, and without the direct comparison with other CTMs summarized in Table 1 (especially WRF-Chem as well stated in the manuscript), it is not appropriate to emphasize the superiority of WRF-GC model. I would like to suggest to focus only on GEOS-Chem Classic model in introduction section. For example, it seems to be one of the significant point in terms of the necessity of fine-scale simulation beyond the Classic nested version. In addition, I am not GEOS-Chem user, but the statement posed me some biases in GEOS-Chem. Especially, some criticism on WRF-Chem model can be avoided as much as possible without appropriate evidences.*

Thank you for the criticism and suggestion. We removed Table 1 to better focus the scope of this work. We also revised large portions of the Introduction and other parts of the text to more objectively describe the advantages of WRF-GC and its comparison with GEOS-Chem Classic. For example:

**P3-4, L80-87, Introduction:**

WRF-GC offers users of WRF-Chem or other regional models the option to use the latest GEOS-Chem chemical module, which is actively developed by a large international user base, well-documented, traceable, benchmarked, and centrally-managed. Through WRF-GC, regional modellers also gain access to the specialty simulations in GEOS-Chem, such as the simulations of mercury (Horowitz et al., 2017; Soerensen et al., 2010) and persistent organic pollutants (Friedman et al., 2013). WRF-GC drives GEOS-Chem with online meteorological fields simulated by WRF, which in turn can be driven by initial and boundary meteorological conditions from many different assimilated

datasets or climate model outputs (Skamarock et al., 2008, 2019). As such, WRF-GC allows GEOS-Chem users to perform high-resolution simulations in both forecast and hindcast modes at any location and time of interest.

**P18, L543-547, Section 6:**

The WRF-GC coupling structure, including the GEOS-Chem column interface and the state conversion module, are extensible and can be adapted to models other than WRF. This opens up possibilities of coupling GEOS-Chem to other weather and Earth System models in an online, modular manner. Using native, out-of-the-box copies of parent models in coupled models reduces maintenance and avoids branching of the parent model code. It also enables the community to more easily transfer developments in the parent models to the coupled model, and vice versa.

*R1.2 Application: Related above mentioned comment, without the evaluation to other models, this study should keep focus on the comparison between GEOS-Chem Classic and developed WRF-GC in terms of the model superiority. In addition, the application is just one case over China and only six days, it should be noted and soften the conclusion derived in this study. In this application, I have following two comments.*

*1. Sections 4.1 and 4.2:*

*GEOS-Chem Classic nested version includes 47 layers with 7 levels in the bottom of 1 km, whereas WRF-GC had 50 layers with 7 levels in the bottom of 1 km. Are these 7 levels (or surface level) identical? Even a few meters differences could cause the modeling performance difference, and I am suspicious the conclusion attributed to the behavior of planetary boundary layer. If the configuration of boundary layer is identical, the height of planetary boundary layer could be one reason as model difference; however, what are other basic meteorological parameters such as wind speed and its direction, temperature, relative humidity, and precipitation (related to wet scavenging)? More investigations on meteorological fields are required in this part.*

Thank you for pointing this out. We revised the text to more objectively describe the comparison between WRF-

GC and GEOS-Chem Classic nested-China simulations of Chinese $PM_{2.5}$. We also expanded the validation of WRF-GC-simulated meteorological variables. We found that the meteorological variables simulated by WRF-GC (with nudging) better represented the spatiotemporal variability of the observed surface temperature, relative humidity, winds, and PBLH, relative to those in the GEOS-FP dataset. We revised the text as follows:

**P12, L369-372, Section 4:**

Our goal was to compare the performance of the two models in simulating Chinese surface $PM_{2.5}$ under their normal mode of operation. To this end, the two simulations were configured as similarly as possible, but there are important innate differences between the two models, as described below.

**P14-15, L438-459, Section 4.2:**

Our analyses above show that the hourly and daily surface $PM_{2.5}$ concentrations simulated by the WRF-GC model were in better agreement with observations than those simulated by the GEOS-Chem Classic nested-China model over Eastern China during January 22 to 27, 2015. We found that this was partially because the WRF-GC model, nudged with surface and upper-air meteorological observations, better represented the pollution meteorology, compared to the GEOS-FP dataset that was used to drive the GEOS-Chem Classic nested-China simulation. Figure S1 shows the average surface air temperature, relative humidity, and 10-m wind speed as simulated by the WRF-GC model and as provided by the GEOS-FP dataset against the observations during January 22 and 27, 2015 at 367 sites over China. The surface air temperature simulated by WRF-GC and those in the GEOS-FP dataset were both in good agreement with the observations over China. However, the relative humidity and wind speeds simulated by WRF-GC were more consistent with the observations, compared to those in the GEOS-FP dataset. Figure S2 assesses the hourly surface air temperature, relative humidity, and near-surface winds simulated by the WRF-GC model and those in the GEOS-FP assimilated meteorological dataset, against hourly surface measurements over China during January 22-27, 2015. For the 34 sites with publicly-available hourly measurements, the meteorological fields simulated by

the WRF-GC were generally more consistent with the measurements.

Figure 5 shows the mean planetary boundary layer height (PBLH) at 08:00 local time (00:00 UTC) and 20:00 local time (12:00 UTC) during January 22 to 27, 2015 in the GEOS-Chem Classic nested-China and the WRF-GC simulations, respectively, and compares them with the rawinsonde observations during this period (Guo et al., 2016). The PBLH in the GEOS-Chem Classic model was taken from the GEOS-FP dataset, whereas the boundary layer height was simulated by WRF in WRF-GC. Compared to the observations, the PBLH in the GEOS-FP dataset were generally biased-low over Eastern China and biased-high over over the mountainous areas in Southwestern China and Western China. This likely was a major reason for the severe overestimation of surface $PM_{2.5}$ concentrations in the GEOS-Chem Classic nested-China simulation over Eastern China. In comparison, the WRF-GC model correctly represented the PBLH over most regions in China, which was critical to its more accurate simulation of surface $PM_{2.5}$ concentrations.

[Figure]

**Figure S1.** Six-day average values of simulated (filled contours) and observed (symbols) 2-m air temperature (upper panel), surface relative humidity (middle panel), and 10-m wind speed (bottom panel) during January 22-27, 2015: (a,c,e) meteorological variables used to drive the GEOS-Chem Classic nested-China simulation (i.e., the GEOS-FP dataset); (b,d,f) meteorological variables simulated by the WRF-GC model. Surface meteorological measurements at 367 sites were obtained from the U.S. National Climate Data Center (https://gis.ncdc.noaa.gov/maps/ncei/cdo/hourly).

[Figure]

**Figure S2.** Assessments of the hourly meteorological variables simulated by the WRF-GC model (red dots) and those used to drive the GEOS-Chem Classic nested-China simulation (i.e., the GEOS-FP dataset, black dots) against hourly measurements at 34 surface sites during January 22-27, 2015: (a) 2-m air temperature, (b) surface relative humidity, (c) 10-m U-wind, and (d) 10-m V-wind. Green, black, and blue dashed lines indicate contours of the normalized centered root-mean-square differences (RMSD), the ratios of simulated versus observed standard deviations, and the Pearson correlation coefficients, respectively. Surface meteorological measurements were obtained from the U.S. National Climate Data Center (https://gis.ncdc.noaa.gov/maps/ncei/cdo/hourly). The 34 sites were selected (out of a total 367 sites) because hourly measurements were publicly-available at these sites.

*R1.3 2. Section 5.1:*

*Again, what is the configuration of vertical layer in this WRF-GC? Without the identical setting of it, the comparison in computational performance is not appropriate. Under the same configuration of both models, the performance differences can be attributed to the importance of online and offline simulation. In addition to the behavior of meteorological fields, how about the importance of online and offline model?*

Thank you for pointing out this lack of clarity. Our intention in Section 5.1 was to compare the computational efficiencies of the two models when they are applied to a similar simulation problem, under their respective typical configuration. We agree that the difference in computational efficiency is influenced by WRF-GC being an online model and GEOS-Chem Classic being an offline model. We revised the manuscript to make the purpose of Section 5.1 more clear, as well as include more details about the model configurations.

**P15, L462-466, Section 5.1:**

We evaluated the computational performance of a WRF-GC simulation and compared it with that of the GEOS-Chem Classic nested-grid simulation of a similar configuration. We configured the WRF-GC and GEOS-Chem Classic nested-grid simulations for the exact same domain (as shown in Figure 2(a)), with the exact same projection and horizontal resolution ($0.25°$ latitude $\times$ $0.3125°$ longitude resolution, $225 \times 161$ atmospheric columns). The GEOS-Chem Classic nested-grid simulation had 47 vertical levels, and the WRF-GC simulation comparably had 50 levels.

**P16, L475-483, Section 5.1:**

We found that the difference in computational efficiency was mainly due to the much faster dynamic and transport calculations in the WRF-GC model relative to the transport calculation in the GEOS-Chem Classic nested-grid model. In WRF-GC, the wall time taken up by the entire WRF (including transport, physics, I/O, and model initiation) was 2462.5 seconds. In the GEOS-Chem Classic nested-grid simulation, 50% (8192 seconds) of the total wall time was used for the transport of tracers, including large-scale advection (6355.7 seconds), convective transport (694.2 seconds), and boundary-layer mixing (1142.5 seconds). As a CTM, the GEOS-Chem Classic read archived meteorological data for the entire domain at 3-model-hour intervals from hard drives using a single computational core, which becomes increasingly burdensome for simulations with more grid boxes. In comparison, WRF-GC calculated meteorology online in node memory and updated the model boundary conditions from hard drives every 6 model-hours.

*Minor comments:*

*R1.4 P6, L175-L179: Does WRF-GC cover both yield model and VBS model for SOA? It is not explicitly stated which is available.*

Thank you for pointing out the omission. Yes, GEOS-Chem has two SOA production options. By default, GEOS-Chem simulates SOA production from volatile organic precursors using simple yields. A more complex scheme that includes the VBS model and additional aqueous productions is also available. WRF-GC supports both options. We revised the text to address this point.

**P6, L166-171, Section 2.2:**

GEOS-Chem has two options to describe the production of SOA, and both options are supported in WRF-GC. By default, SOA is produced irreversibly using simple yields from anthropogenic and biogenic volatile organic precursors (Kim et al., 2015). Alternatively, GEOS-Chem can simulate SOA production via the aging of semi-volatile and intermediate volatility POA using a volatility basis set (VBS) scheme (Robinson et al., 2007; Pye et al., 2010), as well as via the aqueous reactions of oxidation products from isoprene (Marais et al., 2016).

*R1.5 P6, L179-180: What is size bins for sulfate, nitrate, ammonium, black carbon, and POA and SOA? Only dust and sea salt is described here.*

Thank you for pointing out the lack of clarity. We revised the manuscript to more precisely describe the treatment of aerosols.

**P6, L158-166, Section 2.2:**

Aerosol species in GEOS-Chem includes secondary inorganic aerosols (sulfate, nitrate, ammonium), elemental carbon aerosol (EC), primary organic carbon (POC), secondary organic aerosol (SOA), dust, and sea salt. By default, secondary inorganic aerosols, EC, POC, and SOA are simulated as speciated bulk masses. Dust aerosols are represented in 4 size bins (0.1-1.0, 1.0-1.8, 1.8-3.0, and 3.0-6.0 $\mu m$) (Fairlie et al., 2007), while sea salt aerosols are represented in 2 size bins (0.1-0.5 and 0.5-4.0 $\mu m$) (Jaeglé et al., 2011). The thermodynamics of secondary inorganic aerosol are coupled to gas-phase chemistry and computed by the ISORROPIA II module (Park et al., 2004; Fountoukis and Nenes, 2007; Pye et al., 2009). EC and POC are represented in GEOS-Chem as partially hydrophobic and partially hydrophilic, with a conversion timescale from hydrophobic to hydrophilic of 1.2 days (Wang et al., 2014). The organic matter to organic carbon (OM/OC) mass ratio is assumed to be 2.1 for POC by default, with an option to use seasonally and spatially varying OM/OC ratios (Philip et al., 2014).

*R1.6 P7, L203: How can we prepare the initial and boundary conditions for chemical variables? It is not well stated in the current document.*

Thank you for pointing out the issue. Initial and boundary concentrations of chemical species are taken from the GEOS-Chem global model output and interpolated to the horizontal and vertical grids of the WRF-GC model using a modified version of the WRF-Chem preprocessor tool `mozbc`. The initial and boundary conditions are read by WRF

in netCDF format. We have revised the manuscript to clarify.

**P7-8, L213-217, Section 3.1:**

IC/BC for meteorological and chemical variables are prepared by the user in netCDF format and read by WRF. Meteorological IC/BC can be prepared using the WRF pre-processor system (WPS) from datasets available from NCAR's Research Data Archive (`https://rda.ucar.edu`). IC/BC of chemical species concentrations are taken from GEOS-Chem Classic global model outputs and interpolated to the WRF-GC models grids using a modified version of the WRF-Chem preprocessor tool `mozbc` (available along with the WRF-GC code).

*R1.7 P7, L206: What is the "choice of chemical species"? In P6, L163-164, it is only stated "241 chemical species and 981 reactions". Do we need to set chemical species in model simulation?*

*P7, L206: What is the "chemical mechanisms"? In P6, L160-161, we can see "the standard chemical mechanism in GEOS-Chem", but what is other options in GEOSChem?*

Thank you for pointing out the issue. This was an oversight in the wording of the paragraph and we have revised the text to clarify.

**P6, L146-149, Section 2.2:**

Chemical calculations in WRF-GC v1.0 use GEOS-Chem version 12.2.1 (doi:10.5281/zenodo.2580198). The standard chemical mechanism in GEOS-Chem v12.2.1, used by default in WRF-GC, includes detailed $O_x$-$NO_x$-VOC-ozone-halogen-aerosol in the troposphere, as well as the Unified tropospheric-stratospheric chemistry extension (UCX) (Eastham et al., 2014) for stratospheric chemistry and stratosphere-troposphere exchange.

**P6, L151-155, Section 2.2:**

In addition, GEOS-Chem uses the `FlexChem` pre-processor (a wrapper for the Kinetic PreProcessor, KPP, Damian

et al. (2002); Sandu and Sander (2006)) to configure chemical kinetics (Long et al., 2015). This allows users to add or modify gaseous species and reactions to develop custom mechanisms and diagnostic quantities in GEOS-Chem. GEOS-Chem also supports the optional "Tropchem" (troposphere-only chemistry) mechanism, where UCX is disabled and replaced by a parameterized linear chemistry in the stratosphere (McLinden et al., 2000).

**P7-8, L211-213, Section 3.1:**

Chemical options within GEOS-Chem, including the choice of standard or custom chemical mechanisms, emission inventories in HEMCO, and diagnostic quantities to be output, are defined by users in the GEOS-Chem configuration files (`input.geos`, `HEMCO_Config.rc`, and `HISTORY.rc`).

*R1.8 P7, L206: What is the meaning of diagnostics? This wording is ambiguous.*

Thank you for pointing out the lack of clarity. GEOS-Chem refers to all output quantities, including for example chemical concentrations, and production and loss rates, from GEOS-Chem as "diagnostics". We replaced the word with "diagnostic quantities" throughout the text. For example:

**P7-8, L211-213, Section 3.1:**

Chemical options within GEOS-Chem, including the choice of standard or custom chemical mechanisms, emission inventories in HEMCO, and diagnostic quantities to be output, are defined by users in the GEOS-Chem configuration files (`input.geos`, `HEMCO_Config.rc`, and `HISTORY.rc`).

**P8, L225-226, Section 3.1:** At the end of the WRF-GC simulation, WRF outputs all meteorological and chemical diagnostic quantities in WRF's standard format.

*R1.9 P8, L230: This means that WRF-GC only supports the newly established WRF vertical grids, and does*

*not have the choice to old terrain-following grids?*

Thank you for pointing this out. Yes, WRF-GC can only use the hybrid sigma-eta grids, which has become the default option in WRF for versions 4 and higher. We revised the text and the new Table 4 to note this.

**P4, L95-97, Section 2.1:**

WRF uses the Advanced Research WRF (ARW) dynamical solver, which solves fully compressible, Eulerian non-hydrostatic equations on either hybrid sigma-eta (default) or terrain-following vertical coordinates defined by the user.

**P4, L105-106, Section 2.1:**

Table 3 lists the configuration and physical options supported by WRF-GC v1.0. In particular, only the hybrid sigma-eta vertical coordinate is currently supported in WRF-GC.

*R1.10 P10, L310-311: So what is the available lightning NOx emissions in current WRF-GC? It is not available, or using climatorogical value?*

Thank you for pointing out this issue. There are currently no Lightning NOx emissions in WRF-GC. We revised the text to note this.

**P7, L185-186, Section 2.2:**

With the exception of lightning $NO_x$, these meteorology-dependent emissions are supported in WRF-GC v1.0.

**P11, L327-328, Section 3.3.1:**

Lightning $NO_x$ emissions is not yet supported in WRF-GC v1.0 but will be added in a future version.

*R1.11 P11, L317-319 and P11, L328-335: From these paragraphs, I understand that the WRF-GC simulation have the limitations in the selection of various WRF options. This information is necessary to understand the limitation of applicability of WRF-GC model.*

*Therefore, I would like to strongly suggest to put table for the summary of available options in WRF for WRF-GC.*

Thank you for the important suggestion. We added Table 3 to the main text to summarize the WRF options supported in WRF-GC v1.0:

Table 3: List of WRF configuration and physical options supported in WRF-GC v1.0

| Namelist option | Description | Supported value |
|---|---|---|
| **WRF Preprocessing System (WPS)** | | |
| max_dom | Maximum number of domains | 1 |
| map_proj | Map projection | lat-lon; mercator |
| geog_data_res | Static geographical data source | usgs_* |
| **WRF dynamics** | | |
| hybrid_opt | Use hybrid sigma-pressure grid? | 2 (Yes) |
| **WRF physics** | | |
| bl_pbl_physics | Planetary boundary layer | All |
| cu_physics | Cumulus parameterization | 7 (Zhang-McFarlane scheme), 16 (New Tiedtke scheme) |
| mp_physics | Microphysics option | 6 (WRF single-moment 6-class scheme), 8 (New Thompson scheme), 10 (Morrison double-moment scheme) |
| ra_lw_physics | Longwave radiation | 3 (CAM3 scheme), 4 (RRTMG), 5 (New Goddard scheme) |
| ra_sw_physics | Shortwave radiation | 4 (RRTMG shortwave) |
| sf_sfclay_physics | Surface layer | All |
| sf_surface_physics | Land surface | All |
| sf_lake_physics | Lake physics | All |
| sf_urban_physics | Urban surface | All |

*R1.12 P11, L327: For WRF model itself, the land cover can be obtained by both USGS and MODIS, but WRF-GC can be available both datasets? Please explicitly stated for the available land cover information.*

Thank you for pointing out this omission. Only the USGS land cover classification is supported in WRF-GC v1.0. Routines to map the MODIS land cover classification to work with WRF-GC have been reserved in the code and can be implemented in a future WRF-GC version. We have included this information in Table 4, and have revised the text as follows:

**P12, L352-357, Section 3.3.3:**

Dry deposition is calculated in GEOS-Chem using a resistance-in-series scheme (Wesely, 1989; Wang et al., 1998). The land cover data for the simulated domain is read by and used in WRF, but for now WRF-GC only supports the use of the U.S. Geological Survey (USGS) classification. The land cover information is passed to GEOS-Chem, where it is mapped to the land cover classifications of Olson et al. (2001) to assign values of surface roughness and canopy resistance (Wang et al., 1998). The dry deposition velocities are calculated locally using WRF-simulated surface air momentum, sensible heat fluxes, temperature, and solar radiation.

*R1.13 P13, L391 and P25, L460: I understand that this is one test case, but the comparison should not be "preliminary".*

Thank you for your suggestion. We expanded on the analysis of the hourly $PM_{2.5}$ concentrations simulated by the WRF-GC and GEOS-Chem Classic model, as well as the analysis of the meteorological fields in the two models.

**P14-15, L411-459, Section 4.2:**

Figure 2 compares the 6-day average surface $PM_{2.5}$ concentrations during January 22 to 27, 2015 as simulated by WRF-GC and GEOS-Chem Classic, respectively. Also shown are the $PM_{2.5}$ concentrations measured at 578 surface sites, managed by the Ministry of Ecology and Environment of China (www.cnemc.cn). We removed invalid hourly

[revised manuscript text omitted]

Thank you for pointing out this issue. We have carefully revised the wording throughout the text to improve clarity and consistency. For example:

**P14-15, L440-442, Section 4.2:**

We found that this was partially because the WRF-GC model, nudged with meteorological observations, better represented the pollution meteorology, compared to the GEOS-FP dataset that was used to drive the GEOS-Chem Classic nested-China simulation.

**Figure 5 (previously Figure 4) caption:**

Comparison of the simulated (fill contours) and observed (fill symbols) planetary boundary layer heights (PBLH) at 08:00 local time (upper panel) and 20:00 local time (bottom panel) averaged between January 22 and 27, 2015. (a,c) PBLH from the GEOS-FP dataset, which was used to drive the GEOS-Chem Classic nested-China simulation, and (b,d) PBLH simulated by the WRF-GC model.

*R1.15 P14, L424-431: I guess that this WRF-GC model for the discussion of scalability is same model in Section 5.1. If so, this paragraph contains some redundant information. Please clarify and repetition should be avoided.*

Thank you for pointing out this issue. We revised the manuscript for greater clarity and avoided repetition.

**P16, L500, Section 5.2:**

We analyzed the scalability of the WRF-GC model using timing tests of the 48-hour simulation described in

*R1.16 Figure 1: In left corner of "WRF-GC input", there is the statement on "emissions". What is the difference of this emissions and HEMCO?*

Thank you for pointing out this issue. We revised Figure 1 to clarify that the "emission" refers to the emissions input files. HEMCO is the GEOS-Chem module that performs the emission calculation.

**WRF-GC Model (v1.0)**

Figure 1: Architectural overview of the WRF-GC model (v1.0). The WRF-GC Coupler (all parts shown in red) includes interfaces to the two parent models, as well as the state conversion and state management modules. The parent models (shown in grey) are standard codes downloaded from their sources, without any modifications.

*R1.17 Figures 2 and 4: The map line is thinner, so please enhance the map line for readability.*

Thank you for pointing out the issue. We redrew Figures 2 and 4 to improve readability.

[Figure]

Figure 2: Comparison of the simulated (filled contours) 6-day average PM$_{2.5}$ concentrations during Jan 22 to 27, 2015 from (a) the GEOS-Chem Classic nested-China simulation and (b) the WRF-GC nudged simulation. Also shown are the observed 6-day average PM$_{2.5}$ concentrations during this period at 578 surface sites managed by the Ministry of Ecology and Environment of China.

[Figure]

Figure 5: Comparison of the simulated (fill contours) and observed (fill symbols) planetary boundary layer heights (PBLH) at 08:00 local time (upper panel) and 20:00 local time (bottom panel) averaged between Jan 22 and 27, 2015. (a,c) GEOS-Chem Classic nested-China simulation (read from the GEOS-FP dataset), (b,d) WRF-GC simulation.

*R1.18 Figure 6: Gray lines is hard to see. Please change the thickness of lines, or change into black color for readability.*

Thank you for pointing out this issue. We revised the y-axis scale, the color, and the line thickness in Figure 7 to improve readability.

[Figure]

Figure 7: WRF-GC model scalability by processes. Gray lines indicate perfect scalability, i.e. halved computational time for each doubling of processor cores.

*R1.19 Technical comments: L2 in the caption of Figure 3: Need space ("theGEOS-Chem").*

Corrected. Thank you.

**2 Reviewer #2**

*This paper describes how GEOS-Chem has been implemented in the WRF model. The topic and the description of its implementation is appropriate for the GMD audience. There are certain details that need elaboration for future users of the code and I have made several suggestions. In addition, some the statements are clearly biased that overstate the capabilities of the model without actual proof. At times, the text sounds more salesmanship than scientific. Both of these concerns should be relatively easy to address.*

*Major Comments:*

*R2.1 Boundary layer mixing is handled separately by GEOS-Chem. I am a concerned about this approach which is glossed over. Most, but not, all of the meteorology from WRF is used and that point is passed over in other parts of the text. WRF-Chem uses parameters from the WRF boundary layer parameterizations so that the vertical mixing is treated in as similar as possible. In this way, mixing of chemical species occurs within the same boundary layer depth as the meteorology. It is not clear whether vertical mixing in GEOS-Chem and WRF are consistent. If not, it may be possible for GEOSChem to have a deeper or shallower boundary layer than in WRF. If deeper, than more chemistry variables will be transported by free tropospheric air and boundary layer air. The authors should delve into this in more detail to let users know what the approach for vertical mixing actually is and the potential consequences. This applies to the results shown in Figure 4.*

Thank you for pointing this out. In WRF-GC, GEOS-Chem does use the meteorological variables (including boundary layer height and thermodynamic variables) from WRF to calculate boundary-layer mixing. We have revised the text to make this clear.

**P12, L347-350, Section 3.3.2:**

Boundary layer mixing is calculated in GEOS-Chem using a non-local scheme (Holtslag and Boville, 1993; Lin and McElroy, 2010). The boundary layer height, thermodynamic variables, and the vertical level and pressure

information are calculated by WRF and passed to GEOS-Chem through the state conversion module. Again, this methodology is the same as that in the WRF-Chem model.

**P7, L187-191, Section 2.2:** Other physical calculations in GEOS-Chem are coupled to WRF meteorological fields in WRF-GC; we describe the coupling in detail in Section 3.3. Convective transport of chemical species is calculated using a single-plume parameterization (Allen et al., 1996; Wu et al., 2007), which is in turn driven by the cumulus parameterization in WRF. Boundary layer mixing is calculated using a non-local scheme, driven by the WRF-simulated atmospheric instability and boundary layer height (Lin and McElroy, 2010).

*R2.2 At several places in the manuscript, the authors point out the reasons for the advantages of coupling GEOS-Chem within WRF. For the GEOS-Chem community the advantages are obvious. But it is not as clear what the WRF-Chem community gains. The WRF-Chem community would have another chemistry option, but this paper does not explore the types of chemical processes that might be missing (perhaps halogen chemistry) already in the model. The current aerosol treatment is rather simple (is it even "state-of the science"?) and similar in many respects to existing simple aerosol options in WRF-Chem. Users choose particular treatments in a community model, such as WRF, for various reasons such as overall performance (i.e. how well the model represents reality), physics complexity, and computational considerations. The authors have failed to articulate what the advantages are to the WRF community. The future developments beyond v1.0 may change this picture and make that argument more apparent. But if the authors want to frame their arguments in this paper, then a few more concrete points of the advantages to the WRF community are needed.*

Thank you for pointing out this important issue. In our view, the biggest advantage of the WRF-GC code is that it couples native copies of WRF and GEOS-Chem, such that the very active developments in the two parent models by their respectively community can be quickly incorporated into WRF-GC. The use of unmodified copies of the

parent models also allow WRF-GC users to more easily contribute their developments back to the parent models.

We modified the text to reflect these points:

**P3-4, L69-87, Introduction:**

In this work, we developed a new online regional atmospheric chemistry model, WRF-GC, by coupling the WRF meteorology model with the GEOS-Chem chemistry model. Both WRF and GEOS-Chem are open-source and actively developed by the community. We constructed WRF-GC with the following guidelines, in order to best take advantage of new developments in the two parent models and to enhance usability:

1. The coupling structure of WRF-GC should be abstracted from the parent models, and both parent models remain unmodified from their respective sources. In this way, future updates of the parent models can be quickly incorporated into WRF-GC with ease, such that WRF-GC can stay cutting-edge. It also enables WRF-GC users to more easily contribute their developments back to the parent models.

2. The WRF-GC coupled model should scale from conventional computation hardware to massively parallel computation architectures.

3. The WRF-GC coupled model should be easy to install and use, open-source, version-controlled, and well-documented.

WRF-GC offers users of WRF-Chem or other regional models the option to use the latest GEOS-Chem chemical module, which is actively developed by a large international user base, well-documented, traceable, benchmarked, and centrally-managed. Through WRF-GC, regional modellers also gain access to the specialty simulations in GEOS-Chem, such as the simulations of mercury (Horowitz et al., 2017; Soerensen et al., 2010) and persistent organic pollutants (Friedman et al., 2013). WRF-GC drives GEOS-Chem with online meteorological fields simulated by WRF, which in turn can be driven by initial and boundary meteorological conditions from many different assimilated datasets or climate model outputs (Skamarock et al., 2008, 2019). As such, WRF-GC allows GEOS-Chem users to perform high-resolution simulations in both forecast and hindcast modes at any location and time of interest.

**P18, L545-547, Section 6:**

Using native, out-of-the-box copies of parent models in coupled models reduces maintenance and avoids branching of the parent model code. It also enables the community to more easily transfer developments in the parent models to the coupled model, and vice versa.

*R2.3 Specific Comments: Lines 7-8: "is designed to by easy to use, ... extendable, and easy to update" are relative terms and depends on an individual's point of view. This is something I'm struggling with here. "is designed" is probably the key phrase. It is hard for a reviewer to verify the last part of the sentence without using the code itself. WRF itself is a rather complex model, although users with "sufficient" expertise in atmospheric models and computational hardware can learn how to run the model in a short period (i.e. days) of time. Those without "sufficient" expertise, might not describe it as easy to use. Nor am I sure what "extendable" means here. All computational models by their very nature are extendable by modifying code.*

Thank you for your suggestion. We deleted this sentence but added descriptive details to the related statements in the abstract.

**Abstract, L6-10**

WRF-GC uses unmodified copies of WRF and GEOS-Chem from their respective sources; the coupling structure allows future versions of either one of the two parent models to be integrated into WRF-GC with relative ease. Within WRF-GC, the physical and chemical state variables are managed in distributed memory and translated between WRF and GEOS-Chem by the WRF-GC Coupler at runtime.

**Abstract, L15-19**

The WRF-GC model is parallelized across computational cores and scales well on massively parallel architectures. In our tests where the two models were similarly configured, the WRF-GC simulation was three times more efficient than the GEOS-Chem Classic nested-grid simulation, owing to the efficient transport algorithm and the MPI-based parallelization provided by the WRF software framework.

*R2.4 Lines 15-17. I have some concerns regarding the statement on PBL heights, in which I will comment on later in the appropriate section.*

Thank you for your comments. We revised the manuscript and added further analysis of the simulated meteorological fields. Please refer to responses to your comment R2.25.

*R2.5 Lines 18-19: The sentence "Both parent models..." is redundant with an earlier statement. It probably can be deleted and the thought merged with the earlier statement.*

The previous sentence with repeated information was deleted. This sentence was revised as follows.

**Abstract, L15-16**

The WRF-GC model is parallelized across computational cores and scales well on massively parallel architectures.

*R2.6 Lines 27-28: "regional" is used twice in this sentence and is redundant and is self evident.*

Thank you for your suggestion. We revised the text.

**P2, L24-25, Introduction:**

Regional models of atmospheric chemistry simulate the emission, transport, chemical evolution, and removal of

atmospheric constituents over a given domain.

*R2.7 Lines 33-34: "to better serve the public, inform policy makers, and advance science" is a laudable goal. However, I think these types of models are used primarily to "advance science". WRF has been used to provide short-range operational forecasts (e.g. HRRR model), but NOAA is phasing out the use of WRF. WRF-CMAQ has been serving EPA regulatory interest in addition to advancing science, but I am not sure GEOS-Chem serves that purpose.*

Thank you for pointing this out. GEOS-Chem has been used as part of the GEOS system to provide a near real-time forecast at NASA in the GEOS-CF (Composition Forecast) system (`https://gmao.gsfc.nasa.gov/weather\_prediction/GEOS-CF/`). Here, we revised the text to better represent the potential applications of WRF-GC.

**P2, L29-30, Introduction:**

We present here the development of a new regional atmospheric chemistry model, WRF-GC, specifically designed to allow easy updates and be computationally efficient, for use in research and operation applications.

*R2.8 Lines 73-86: The authors state that online models are more difficult to keep up to date than offline models. I disagree. There are many factors that contribute to how quickly and the frequency of updates made in models (some described by the authors) that have nothing to do with whether the model is online or offline. The authors first mention resources used to provide benchmarking, validating, and documentation, which is probably a good thing for both offline and online models. This process does take longer for more complex models, but offline models can be complex too. It is also not a good idea to translate research findings into publicly available without due diligence. Some times new scientific findings are proven to be incorrect and/or the findings are based on a limited case study; therefore, it is not wise or appropriate for use by a broad community. The authors second*

*point on expertise residing in different communities. I agree this is an issue, but this is largely governed by how the organization wishes to develop the code, and not whether the model is offline or online. In the case of WRF, it has always been a community model so that contributions originate from various organizations and universities. Nor has NCAR insisted that all physics schemes be compatible with each other, and sometimes there are good reasons why some physics schemes should not be made compatible. Had NCAR decided to be the sole developer, the code would no doubt be more streamlined but would probably lack scientific options many users want. It seems strange to me that the authors are arguing that bring communities together introduces problems, when the purpose of WRF-GC is yet another iteration of bringing very different communities together. The authors correctly point out that not everything in WRF-Chem (and WRF for that matter) is compatible, but it is not clear whether the same holds for the various treatments in GEOS-Chem.*

Thank you for the very important comment. We modified the text to better frame the advantages of the WRF-GC model. In our view, the biggest advantage of the WRF-GC code is that it couples native copies of WRF and GEOS-Chem, such that the very active developments in the two parent models by their respectively community can be quickly incorporated into WRF-GC. The use of unmodified copies of the parent models also allow WRF-GC users to more easily contribute their developments back to the parent models.

We modified the text to reflect these points:

**P3, P63-68, Introduction:**

However, keeping the representation of atmospheric processes up-to-date is potentially more challenging for online models than it is for offline models. One of the reasons for this is that the interactions between the chemical and meteorological modules are hard-wired in some online models, such that updating either module requires considerable effort. For the same reason, if users make improvements to the chemical or meteorological processes in the online model, those improvements may be relatively difficult to propagate to the broader community. This may lead to the model diverging into different branches, and users may be forced to work with stale, branched versions

of the code.

**P3, L69-76, Introduction:**

In this work, we developed a new online regional atmospheric chemistry model, WRF-GC, by coupling the WRF meteorology model with the GEOS-Chem chemistry model. Both WRF and GEOS-Chem are open-source and actively developed by the community. We constructed WRF-GC with the following guidelines, in order to best take advantage of new developments in the two parent models and to enhance usability:

1. The coupling structure of WRF-GC should be abstracted from the parent models, and both parent models remain unmodified from their respective sources. In this way, future updates of the parent models can be quickly incorporated into WRF-GC with ease, such that WRF-GC can stay cutting-edge. It also enables WRF-GC users to more easily contribute their developments back to the parent models.

**P18, L545-547, Section 6:**

Using native, out-of-the-box copies of parent models in coupled models reduces maintenance and avoids branching of the parent model code. It also enables the community to more easily transfer developments in the parent models to the coupled model, and vice versa.

*R2.9 Lines 98-98. This sentence is kind of a put-down to WRF (in the context of this paragraph) and other models listed in Table 1. As with another statement in the introduction, this depends on one's point-of-view and difficult to prove. What is "state-of-the-science" anyway? This overused phrase is almost meaningless at this point. For examples, some chemistry and aerosol disciplines are changing very rapidly (weekly) and it is very difficult to argue which model has the most "up-to-date" science. I suggest the authors rephrase this sentence to be a bit more fair and unbiased.*

Thank you for your suggestion. We have revised the manuscript according to your comments.

**P3-4, L80-83, Introduction:**

WRF-GC offers users of WRF-Chem or other regional models the option to use the latest GEOS-Chem chemical module, which is actively developed by a large international user base, well-documented, traceable, benchmarked, and centrally-managed. Through WRF-GC, regional modellers also gain access to the specialty simulations in GEOS-Chem, such as the simulations of mercury (Horowitz et al., 2017; Soerensen et al., 2010) and persistent organic pollutants (Friedman et al., 2013).

*R2.10 Line 100: The authors state that that GEOS-Chem is driven by the meteorological fields simulated by WRF. But after reading material later in the paper, this may not be entirely true. It seems that GEOS-Chem still uses its own turbulent vertical mixing which could be different from WRF. I have some other comments on this point later in the paper.*

Thank you for pointing this out. In WRF-GC, GEOS-Chem does calculate turbulent mixing in the PBL, but the calculation is driven by WRF-simulated meteorology. This methodology is same as the methodology in WRF-Chem. We have revised section 3.3.2 regarding vertical mixing.

**P12, L347-350, Section 3.3.2:**

Boundary layer mixing is calculated in GEOS-Chem using a non-local scheme (Holtslag and Boville, 1993; Lin and McElroy, 2010). The boundary layer height, thermodynamic variables, and the vertical level and pressure information are calculated by WRF and passed to GEOS-Chem through the state conversion module. Again, this methodology is the same as that in the WRF-Chem model.

**P7, L187-191, Section 2.2:** Other physical calculations in GEOS-Chem are coupled to WRF meteorological fields in WRF-GC; we describe the coupling in detail in Section 3.3. Convective transport of chemical species is calculated using a single-plume parameterization (Allen et al., 1996; Wu et al., 2007), which is in turn driven by

the cumulus parameterization in WRF. Boundary layer mixing is calculated using a non-local scheme, driven by the WRF-simulated atmospheric instability and boundary layer height (Lin and McElroy, 2010).

*R2.11 Lines 168-169: Based on this sentence, it sounds like the bulk treatment is similar to GOCART. But the last sentence in the paragraph implies a modal treatment for dust and sea-salt. So is dust and sea-salt in bulk bins or actually prescribed using a size distribution? Just need to be consistent in the text here.*

Thank you for pointing this out. We revised the text to include more detail:

**P6, L158-161, Section 2.2:**

Aerosol species in GEOS-Chem includes secondary inorganic aerosols (sulfate, nitrate, ammonium), elemental carbon aerosol (EC), primary organic carbon (POC), secondary organic aerosol (SOA), dust, and sea salt. By default, secondary inorganic aerosols, EC, POC, and SOA are simulated as speciated bulk masses. Dust aerosols are represented in 4 size bins (0.1-1.0, 1.0-1.8, 1.8-3.0, and 3.0-6.0 $\mu m$) (Fairlie et al., 2007), while sea salt aerosols are represented in 2 size bins (0.1-0.5 and 0.5-4.0 $\mu m$) (Jaeglé et al., 2011).

*R2.12 Lines 169-172: These two sentences might be better at the end of the paragraph. It would be better to have all the discussion on the present aerosol model together, rather than broken up with what is not in the code in the middle.*

Thank you for your suggestion. We revised the paragraph to improve clarity:

**P6, L158-174, Section 2.2:**

Aerosol species in GEOS-Chem includes secondary inorganic aerosols (sulfate, nitrate, ammonium), elemental carbon aerosol (EC), primary organic carbon (POC), secondary organic aerosol (SOA), dust, and sea salt. By default, secondary inorganic aerosols, EC, POC, and SOA are simulated as speciated bulk masses. Dust aerosols are

represented in 4 size bins (0.1-1.0, 1.0-1.8, 1.8-3.0, and 3.0-6.0 $\mu m$) (Fairlie et al., 2007), while sea salt aerosols are represented in 2 size bins (0.1-0.5 and 0.5-4.0 $\mu m$) (Jaeglé et al., 2011). The thermodynamics of secondary inorganic aerosol are coupled to gas-phase chemistry and computed by the ISORROPIA II module (Park et al., 2004; Fountoukis and Nenes, 2007; Pye et al., 2009). EC and POC are represented in GEOS-Chem as partially hydrophobic and partially hydrophilic, with a conversion timescale from hydrophobic to hydrophilic of 1.2 days (Wang et al., 2014). The organic matter to organic carbon (OM/OC) mass ratio is assumed to be 2.1 for POC by default, with an option to use seasonally and spatially varying OM/OC ratios (Philip et al., 2014). GEOS-Chem has two options to describe the production of SOA, and both options are supported in WRF-GC. By default, SOA is produced irreversibly using simple yields from anthropogenic and biogenic volatile organic precursors (Kim et al., 2015). Alternatively, GEOS-Chem can simulate SOA production via the aging of semi-volatile and intermediate volatility organic precursors using a volatility basis set (VBS) scheme (Robinson et al., 2007; Pye et al., 2010), as well as via the aqueous reactions of the oxidation products from isoprene (Marais et al., 2016). The GEOS-Chem model also has the option of simulating detailed, size-dependent aerosol microphysics using the TwO-Moment Aerosol Sectional microphysics (TOMAS) module (Kodros and Pierce, 2017) or the Advanced Particle Microphysics (APM) module (Yu and Luo, 2009), but these two modules are not yet supported in WRF-GC.

*R2.13 Lines 181-185: I think some discussion is needed regarding higher-resolution emission datasets. GEOS-Chem has been used traditionally at global scales and there are several global emission datasets available. But if the point of WRF-GC is to run at higher resolution, this would be defeated in part by an emission inventory that is much coarser than what could be simulated by the model. What is the strategy for that? Will other inventories used by EPA and the WRF-Chem community be used? Or are users expected to generate emissions on their own? Also, emissions are discussed in more detail later in Section 3.3.1. So I am wondering if it is even necessary to talk about emissions here. This material can be merged into Section 3.3.1.*

Thank you for pointing out the lack of clarity. We revised the text to first describe how HEMCO calculates emissions in GEOS-Chem and the default inventories in Section 3.3.1. We then describe how WRF-GC can choose to use the existing inventories in HEMCO or prepare their own in Section 3.3.1.

**P6-7, L175-186, Section 2.2:**

Emissions of chemical species in WRF-GC are calculated using the HEMCO emissions component in GEOS-Chem (Keller et al., 2014). HEMCO allows users to select emission inventories from the HEMCO data directory or add their own inventories, and interpolate the emission fluxes to the model domain and resolution at runtime. The HEMCO data directory currently includes more than 20 global and regional emission inventories, mostly at their respective native resolutions (`http://wiki.seas.harvard.edu/geos-chem/index.php/HEMCO_data_directories`). By default, the Community Emissions Data System (CEDS) inventory ($0.5 \times 0.5$ resolution, monthly) (Hoesly et al., 2018) is used for most of the world; over Asia and the U.S., the CEDS is superseded by the MIX inventory ($0.25 \times 0.25$ native resolution, monthly, Li et al. (2017b)) and the 2011 National Emission Inventory (NEI 2011) (0.1 km $\times$ 0.1 km native resolution, hourly, U.S. Environmental Protection Agency (2014)), respectively. HEMCO also has extensions to compute emissions with meteorological dependencies, such as the emissions of biogenic species (Guenther et al., 2012), soil $NO_x$ (Hudman et al., 2012), lightning $NO_x$, sea salt (Gong, 2003), and dust (Zender et al., 2003). With the exception of lightning $NO_x$, these meteorology-dependent emissions are supported in WRF-GC v1.0. Further details about the use of HEMCO in WRF-GC is given in Section 3.3.1.

**P11, L318-328, section 3.3.1**

Chemical emissions in WRF-GC are calculated online by the HEMCO module in GEOS-Chem (Keller et al., 2014) and configured in `HEMCO_Config.rc`. HEMCO and its data directory are updated as part of the GEOS-Chem model and remain unmodified in WRF-GC. Users can choose to use one or combine several of the emission inventories already in the HEMCO data directory (Section 2.2). Some of the inventories currently available in the HEMCO data directory may not be of sufficiently fine resolution to support the high-resolution WRF-GC simulations.

In that case, users can prepare their own emission input files in netCDF format at arbitrary spatiotemporal resolutions, and HEMCO will interpolate them to the WRF-GC model domain and resolution at runtime. HEMCO also allow users to specify scale factors and diurnal/weekly/monthly variation profiles in `HEMCO_Config.rc` to be applied to the emission fluxes at runtime. WRF-GC calls HEMCO to compute meteorology-dependent emissions online using WRF-simulated meteorology. These currently include the emissions of biogenic species (Guenther et al., 2012), soil $NO_x$ (Hudman et al., 2012), sea salt (Gong, 2003), and dust (Zender et al., 2003). Lightning $NO_x$ emissions is not yet supported in WRF-GC v1.0 but will be added in a future version.

*R2.14 Line 198: I am not sure what "greater modularity" means here. WRF-Chem is structured in a way to have modularity for important chemistry and aerosol processes. For example, there are several option for online emissions, deposition, etc. The modularity also permits users to add their own treatments if they wish. In some cases, there are treatments that can be used for multiple chemistry or aerosol options. For example, wet scavenging can be handled similarly for MADE-SORGAM (modal aerosols) and MOSAIC (sectional aerosols). But I doubt if the modules in GEOS-Chem can be used or accessed by other treatments in the code, given the structure on how it was implemented. So, if the authors want to use the phrase "greater modularity" some description is need to know exactly what this means.*

Thank you for pointing out this issue. This sentence has been deleted.

*R2.15 Line 206: What is the "WRF-to-chemistry interface". Is GEOS-Chem handled in the registry.chem file, much like the other chemistry options? Given Figure 1, I think this will be discussed later and it looks like the material is in Section 3.2.2. I just found this a bit vague at this point.*

Thank you for your suggestion. We revised the text here and also in Sections 3.2.2 and 3.2.3 to improve clarity.

**P7, L208-210, Section 3.1:**

Users also "turn on" GEOS-Chem in WRF-GC by specifying `chem_opt = 233` in `namelist.input`, similar to the way that users specify the chemical mechanism in WRF-Chem. GEOS-Chem is initialized by the WRF model using the WRF-to-chemistry interface described in Section 3.2.3.

**P9, L266-268, Section 3.2.2:**

When the user sets the environment variable `WRF_CHEM` to 1 in the WRF compile script, WRF reads a registry file (`registry.chem`) containing the GEOS-Chem chemical species information (duplicated from `input.geos`) and builds these species into the WRF model framework.

**P9, L275-279, Section 3.2.3:**

In WRF-Chem, WRF calls its interface to chemistry, `chem_driver`, which then calls each individual chemical processes. We abstracted this `chem_driver` interface by removing direct calls to chemical processes. Instead, our `chem_driver` calls the WRF-GC state conversion module (`WRFGC_Convert_State_Mod`) and the GEOS-Chem column interface (`GIGC_Chunk_Run`) to perform chemical calculations. We also modified `chemics_init` to initializes GEOS-Chem through the column interface `GIGC_Chunk_Init`.

*R2.16 Line 213: See the first major comment above.*

Thank you for your suggestion. We clarified the treatment of boundary-layer mixing.

**P12, L347-350, Section 3.3.2**

Boundary layer mixing is calculated in GEOS-Chem using a non-local scheme (Holtslag and Boville, 1993; Lin and McElroy, 2010). The boundary layer height, thermodynamic variables, and the vertical level and pressure

information are calculated by WRF and passed to GEOS-Chem through the state conversion module. Again, this methodology is the same as that in the WRF-Chem model.

*R2.17 Line 220: In Section 3.2, the text implies that GEOS-chem is compiled and run on top of the host WRF model. The text implies but does not explicitly state that the addition of GEOS-Chem has not impaired the use of WRF-Chem. One of the rules in the WRF community is that any new additions must be shown to not "break" other parts of the code. So have the authors performed tests with the new code to ensure that other chemistry options produce the same results went the code is compiled with GEOSchem?*

Thank you for pointing this out. At the time WRF-GC started development, the workflow for installing WRF-Chem was that the user downloaded the WRF model in `WRFV3/`, and a separate WRF-Chem zip file which contained the subdirectory to be placed in `WRFV3/chem/`. So, at the time, the WRF-GC development did not "break" any part of the WRF model.

Two years later the landscape has changed, as the latest WRF version now bundles the chemistry routines with its distribution. Our implementation of WRF-GC still has not "broken" the "WRF" part of the latest WRF model, but at present the chemical routines of WRF-Chem cannot work alongside the GEOS-Chem chemical module in a single WRF-GC top directory.

We have revised the manuscript to clarify this point.

**P9, L255-260, section 3.2.2**

WRF-GC is installed by downloading the parent models, WRF and GEOS-Chem, and the WRF-GC Coupler, directly from their respective software repositories. The WRF model is installed in a top-level directory, while the WRF-GC Coupler and GEOS-Chem are installed under the `chem/` sub-directory, where the chemistry routines for WRF-Chem originally reside. An unmodified copy of the GEOS-Chem code is installed in the `chem/gc/` sub-directory, and a set of sample GEOS-Chem configuration files is in `chem/config/`. The WRF meteorology model

remains unmodified in WRF-GC, but at present the chemical routines of WRF-Chem cannot work alongside GEOS-Chem under a single WRF-GC top directory.

*R2.18 Lines 312-335: As the authors imply, subgrid and removal processes, will depend on both "resolved" and "unresolved" clouds. In contrast with global models, simulations at fine enough resolutions may be best run with any "unresolved" cloud parameterizations. Ideally, a cumulus parameterization would have a progressively smaller and smaller impact at higher and higher resolutions. The subgrid and removal processes depend on the current behavior of what is in WRF. It would be useful to provide some discussion for users regarding the implications of these assumptions. Since this paper describes a new modeling framework, it would have been interesting to see some simulations at coarse and fine resolutions to demonstrate the differences on the vertical transport and wet scavenging. WRF-GC users will be able to run at smaller grid spacings, which is a good thing, but will they understand the subtleties of these assumptions that are not normally encountered at global scales?*

Thank you for your suggestion. We added more detailed descriptions about the cumulus parameterization schemes currently implemented in WRF-GC, as well as recommendations for the choice of cumulus parameterization at different horizontal resolution.

**P11, L338-346, Section 3.3.2:**

In addition, the users should consider the horizontal resolution of the model when choosing which cumulus parameterization to use. The New Tiedtke scheme and the Zhang-McFarlane schemes are generally recommended for use in simulations at horizontal resolutions larger than 10 km (Skamarock et al., 2008; Arakawa and Jung, 2011). At horizontal resolutions between 2 to 10 km, the so-called "convective grey zone" (Jeworrek et al., 2019), the use of the Grell-Freitas scheme is recommended for the WRF model (Grell and Freitas, 2014), as it allows subsidence to spread to neighboring columns; this option will be implemented in a future WRF-GC version. At horizontal resolutions finer than 2 km, it is assumed that convections are resolved and cumulus parameterizations should not be

used (Grell and Freitas, 2014; Jeworrek et al., 2019). The scale-dependence of cumulus parameterizations and their impacts on convective mixing of chemical species is an active area of research, which we will explore in the future using WRF-GC.

*R2.19 Lines 323-325: Same comment as before regarding vertical mixing. This is a different treatment than in WRF. So there could be mis-matches in how boundary layer mixing, and it impact the vertical extent of boundary layer mixing, between WRF (which is used to transport chemical species) and GEOS-Chem.*

Thank you for pointing out the lack of clarity. In WRF-GC, GEOS-Chem calculate boundary-layer mixing using meteorological fields from WRF. We revised the manuscript to clarify.

**P12, L347-350, Section 3.3.2**

Boundary layer mixing is calculated in GEOS-Chem using a non-local scheme (Holtslag and Boville, 1993; Lin and McElroy, 2010). The boundary layer height, thermodynamic variables, and the vertical level and pressure information are calculated by WRF and passed to GEOS-Chem through the state conversion module. Again, this methodology is the same as that in the WRF-Chem model.

*R2.20 Lines 350-359: The authors should include the meteorology simulation time step and chemical time step. What is not mentioned in the discussion of the model's implementation, but alluded to in the conclusion, is that a different chemical time step can be used (which is similar to WRFChem). In this way, transport of chemical species are done at the meteorological time step. Chemistry is usually simulated at a coarser time step to save computer time, but some care is needed since chemical time steps that are too large could introduce uncertainties in the predictions – especially at higher spatial resolutions. Since the paper is about WRF-GC and designed to work at higher spatial resolution, some discussion is needed in the best practice for the two types of time steps. Ideally, they should be the same for consistency.*

Thank you for pointing out this omission. We specified the time steps in the two simulations as suggested.

**P13, L380-381, 393, Section 4.1:**

[For GEOS-Chem Classic nested-China simulation:] The dynamic time step and the external chemistry time step were 5 minutes and 10 minutes, respectively.

[For WRF-GC simulation:] The dynamic time step and the external chemistry time step were 2 minutes and 10 minutes, respectively.

We also added descriptions about the chemical time step in WRF-GC:

**P11, L311-316, Section 3.3:**

The dynamic and chemical time steps are specified by the user in the WRF configuration file `namelist.input`. The dynamic time step is constrained by the Courant-Friedrichs-Lewy stability criterion and should be short for high-resolution simulations. WRF-Chem recommends that the chemical time step be set the same as the dynamic time step as best practice (Peckham et al., 2017). Because GEOS-Chem uses a Rosenbrock solver, which adapts its internal chemical time step to the stiffness of the chemical mechanism, a larger chemical time step may be used. However, it is recommended that the results be compared to a control simulation with the chemical time step set to the dynamic time step (Peckham et al., 2017).

*R2.21 Line 353: The authors use FNL for the meteorological initial and boundary conditions. To be more consistent with the nested GEOS-Chem, why not use the meteorological conditions from that model?*

Thank you for pointing this out. The GEOS-Chem Classic nested model uses the GEOS-FP meteorological dataset (reanalysis data), but WRF currently does not support the use of GEOS-FP as meteorological initial and

boundary conditions. We choose FNL as a representative meteorological dataset, since it is what many WRF-GC

users would choose to use, and nudged WRF with meteorological observations. We added this point in the text.

**P13, L385-387, Section 4.1:**

The WRF model does not have the option of using the GEOS-FP dataset for meteorological IC/BC. Instead, we

used the NCEP FNL dataset (doi:10.5065/D6M043C6) at $1° \times 1°$ resolution as IC/BC for WRF-GC; the FNL dataset

was interpolated to WRF vertical levels and updated every 6 hours.

*R2.22 Lines 356-359: It sounds like the authors are using the obs-nudging capability in WRF. Would be useful*

*to cite a reference on that. What is not clear in the previous paragraph, for readers not familiar with GEOS-Chem,*

*that the meteorological fields are prescribed analyses. Thus nudging in WRF would be appropriate to make the*

*meteorology in the two simulations more compatible. The authors should be more specific on these points.*

Thanks for your suggestion. We added citations to the nudging algorithms in Section 2.1 where the functionalities

of WRF is described. We also revised the text in Section 4.1 to describe the nudging and its effect on our test

simulation.

**P4, L100-102, Section 2.1:**

WRF supports grid-, spectral-, and observational-nudging (Liu et al., 2005, 2006; Stauffer and Seaman, 1990,

1994). This allows the WRF model to produce meteorological outputs that mimic assimilated meteorological fields

for use in air quality hindcasts.

**P13, L387-391, Section 4.1:**

In addition, we nudged the WRF-simulated meteorological fields with surface (every 3 hours) and upper air (every

6 hours) observations of temperature, specific humidity, and winds from the NCEP ADP Global Surface/Upper

Air Observational Weather Database (doi:10.5065/39C5-Z211). This mimicked the effect of meteorological data

assimilation and allowed the WRF-simulated meteorology to stay close to the observed states of the atmosphere.

*R2.23 Line 363-364: Do the anthropogenic emissions vary diurnally? This seems to be an important point in simulating diurnal and peak concentrations. Here the authors only show a 6-day average. Perhaps this is okay for demonstrate WRF-GC compared to GEOS-Chem, and this distinction should be made.*

Thank you for your suggestion. We added description about the weekly/diurnal variation of the anthropogenic emissions. We also expanded the diagnosis of simulated hourly PM$_{2.5}$ concentrations against observations.

**P13, L397-400, Section 4.1:**

Monthly mean anthropogenic emissions from China were from the Multi-resolution Emission Inventory for China (MEIC, Li et al. (2014)) at $0.25° \times 0.25°$ horizontal resolution. The MEIC inventory was updated for the year 2015 and included emissions from power generation, industry, transportation, and residential activities. Sector-specific weekly and diurnal variation from the MEIC inventory were applied (Li et al., 2017a).

**P14, L429-437, Section 4.2:**

Figure 4 shows the Taylor diagrams of the hourly PM$_{2.5}$ concentrations simulated by the two models at 48 major eastern Chinese cities, including 13 cities in the Beijing-Tianjin-Hebei (BTH) area, 22 cities in the Yangtze River Delta (YRD) area, and 13 other major cities. The Taylor diagram (Taylor, 2001) evaluates the simulated time series of PM$_{2.5}$ against the observations, using the Pearson correlation coefficients, the ratio between the simulated and observed standard deviations ($\frac{\sigma_{sim}}{\sigma_{obs}}$), and the normalized root-mean-square differences (RMSDs) as metrics. Prox-imity to the point "1" on the X-axis in Figure 4 indicates that the simulation accurately reproduced both the mean concentration and the temporal variability of the observations. For most cities in the BTH area and for most of the other 13 major cities, the hourly PM$_{2.5}$ concentrations simulated by WRF-GC showed smaller RMSDs and higher

correlation coefficients against observations, compared to those in the GEOS-Chem Classic nested-china simulation.

In the YRD area, the performances of the two models were similar.

[Figure]

Figure 4: Taylor diagrams of PM$_{2.5}$ concentrations during Jan 22 to 27, 2015 from the GEOS-Chem Classic nested-China simulation (black) and WRF-GC nudged simulation (red) in (a) the Beijing-Tianjin-Hebei Region (BTH), (b) the Yangtze River Delta Region (YRD) and (c) major megacities in China.

*R2.24 Lines 394: Are these boundary layer heights the same as predicted by WRF? Or different calculations in the GEOS-Chem modules?*

Thank you for pointing this out. In WRF-GC, the boundary layer height is calculated by the WRF model and passed to GEOS-Chem. We have revised the manuscript to resolve this ambiguity.

**P15, L451-455, Section 4.2:**

Figure 5 shows the mean planetary boundary layer height (PBLH) at 08:00 local time (00:00 UTC) and 20:00 local time (12:00 UTC) during January 22 to 27, 2015 in the GEOS-Chem Classic model and in the WRF-GC model, respectively, and compares them with the rawinsonde observations during this period (Guo et al., 2016). The PBLH in the GEOS-Chem Classic model was taken from the GEOS-FP dataset, whereas the boundary layer height was simulated by WRF in WRF-GC.

*R2.25 Lines 397-402: The 6-day averaging may be hiding some day-to-day variations when trying to assess the cause of the positive PM2.5 bias. While it is very likely that the boundary layer height issue is contributing to that, there are likely other differences in the meteorology that could also be attributing factors. The authors should at least acknowledge that if they do not wish to pursue a more detailed analysis of the simulations. I assume the same boundary layer scheme is being used in both models, but the differences arise in how that scheme is driven. I presume in GEOS-Chem the meteorology is from the large-scale analyses. Whereas, in WRF the prognostic land-surface model will be controlling the evolution of surface temperature and fluxes that will drive a boundary layer parameterization. So, some additional discussion as to why these differences in the boundary layer height would be useful.*

Thanks for your suggestions. We expanded the validation of WRF-GC-simulated meteorological variables. We found that the meteorological variables simulated by WRF-GC (with nudging) better represented the spatiotemporal

variability of the observed surface temperature, relative humidity, winds, and PBLH, relative to those in the GEOS-FP dataset. We revised the text as follows:

**P14-15, L438-459, Section 4.2:**

Our analyses above show that the hourly and daily surface $PM_{2.5}$ concentrations simulated by the WRF-GC model were in better agreement with observations than those simulated by the GEOS-Chem Classic nested-China model over Eastern China during January 22 to 27, 2015. We found that this was partially because the WRF-GC model, nudged with surface and upper-air meteorological observations, better represented the pollution meteorology, compared to the GEOS-FP dataset that was used to drive the GEOS-Chem Classic nested-China simulation. Figure S1 shows the average surface air temperature, relative humidity, and 10-m wind speed as simulated by the WRF-GC model and as provided by the GEOS-FP dataset against the observations during January 22 and 27, 2015 at 367 sites over China. The surface air temperature simulated by WRF-GC and those in the GEOS-FP dataset were both in good agreement with the observations over China. However, the relative humidity and wind speeds simulated by WRF-GC were more consistent with the observations, compared to those in the GEOS-FP dataset. Figure S2 assesses the hourly surface air temperature, relative humidity, and near-surface winds simulated by the WRF-GC model and those in the GEOS-FP assimilated meteorological dataset, against hourly surface measurements over China during January 22-27, 2015. For the 34 sites with publicly-available hourly measurements, the meteorological fields simulated by the WRF-GC were generally more consistent with the measurements.

Figure 5 shows the mean planetary boundary layer height (PBLH) at 08:00 local time (00:00 UTC) and 20:00 local time (12:00 UTC) during January 22 to 27, 2015 in the GEOS-Chem Classic nested-China and the WRF-GC simulations, respectively, and compares them with the rawinsonde observations during this period (Guo et al., 2016). The PBLH in the GEOS-Chem Classic model was taken from the GEOS-FP dataset, whereas the boundary layer height was simulated by WRF in WRF-GC. Compared to the observations, the PBLH in the GEOS-FP dataset were generally biased-low over Eastern China and biased-high over over the mountainous areas in Southwestern China

and Western China. This likely was a major reason for the severe overestimation of surface $PM_{2.5}$ concentrations in the GEOS-Chem Classic nested-China simulation over Eastern China. In comparison, the WRF-GC model correctly represented the PBLH over most regions in China, which was critical to its more accurate simulation of surface $PM_{2.5}$ concentrations.

[Figure]

**Figure S1.** Six-day average values of simulated (filled contours) and observed (symbols) 2-m air temperature (upper panel), surface relative humidity (middle panel), and 10-m wind speed (bottom panel) during January 22-27, 2015: (a,c,e) meteorological variables used to drive the GEOS-Chem Classic nested-China simulation (i.e., the GEOS-FP dataset); (b,d,f) meteorological variables simulated by the WRF-GC model. Surface meteorological measurements at 367 sites were obtained from the U.S. National Climate Data Center (https://gis.ncdc.noaa.gov/maps/ncei/cdo/hourly).

[Figure]

**Figure S2.** Assessments of the hourly meteorological variables simulated by the WRF-GC model (red dots) and those used to drive the GEOS-Chem Classic nested-China simulation (i.e., the GEOS-FP dataset, black dots) against hourly measurements at 34 surface sites during January 22-27, 2015: (a) 2-m air temperature, (b) surface relative humidity, (c) 10-m U-wind, and (d) 10-m V-wind. Green, black, and blue dashed lines indicate contours of the normalized centered root-mean-square differences (RMSD), the ratios of simulated versus observed standard deviations, and the Pearson correlation coefficients, respectively. Surface meteorological measurements were obtained from the U.S. National Climate Data Center (https://gis.ncdc.noaa.gov/maps/ncei/cdo/hourly). The 34 sites were selected (out of a total 367 sites) because hourly measurements were publicly-available at these sites.

*R2.26 Line 417: Eliminating the need to read archived meteorology becomes even more important at high resolution offline approaches as more and more time is required for I/O.*

Thank you for the suggestion. We have revised the text as follows.

**P16, L480-482, Section 5.1:**

As a CTM, the GEOS-Chem Classic read archived meteorological data for the entire domain at 3-model-hour intervals from hard drives using a single computational core, which becomes increasingly burdensome for simulations with more grid boxes.

*R2.27 Line 418: I do not fully understand what reading from disks means. Do the authors mean the analyses are on some long-term storage device (tapes?) in which reading is slower than conventional hard-drives? Would a fairer timing test include having meteorological fields in a place with faster access?*

Thank you for pointing this out. We meant that GEOS-Chem read meteorological data from hard drives. We used the exact same computational hardware with an Ethernet-connected file system for both WRF-GC and GEOS-Chem Classic for the computational performance assessment. We revise the manuscript to resolve this confusion.

**P15, L471-472, section 5.1:**

Both simulations were executed on the same single-node hardware with 32 Intel Broadwell physical cores and an Ethernet-connected hard disk array.

**P16, L480-482, Section 5.1:**

As a CTM, the GEOS-Chem Classic read archived meteorological data for the entire domain at 3-model-hour intervals from hard drives using a single computational core, which becomes increasingly burdensome for simulations with more grid boxes.

*R2.28 Line 432: Figure 6 is difficult to interpret because of the scale on the y-axis. Ideally if one doubled the number of cores, one would want the wall clock time to be reduced by a factor of 2. It is hard to see this with the current scaling. To me it looks like the total performance does not improve much greater than 100 cores. In addition to "fragmentation" the authors mention, the most CPU time in WRF is due to the advection of species. The more chemical species there are the slower the code is. The authors mention 241 species on line 163 (presumably trace gas), but there would be aerosol species on top of that. This is more than other options in WRF-Chem, but less than others. In the conclusion the paper states that the WRF-chem community would benefit from GEOS-Chem, but it is not clear what the computational cost would be compared to other approaches. A large fraction of the users in the WRF community chose "simple" chemistry and aerosol schemes to save computational cost. While the authors may not wish to perform identical simulations with different chemical options to benchmark GEOS-Chem with others, the least they could do is compare the number of species to establish some sort of computational level of complexity.*

Thank you for the excellent suggestion. We revised Figure 6 (now Figure 7) to a log-log plot and modified the discussion on the computational performance. In addition, we added a scalability test of WRF-GC on the cloud using the Amazon Web Services cloud.

**P17, L518-522, Section 5.2:**

The scalability test results are shown in Figure S3. In this massively parallel environment, WRF-GC scaled well up to 1728 cores, with the chemical module scaling well up to 2304 cores. The WRF-GC Coupler took less than 0.2% of the total computational time in this simulation and scaled perfectly up to 4608 cores. The deployability of WRF-GC on the cloud will enhance WRF-GC's accessibility to new users by saving them the investment in hardware purchases and the effort in downloading and hosting large input datasets locally.

We also added a paragraph to compare the performance of WRF-Chem and WRF-GC, while specifying the dif-

[Figure]

[Figure]

**Figure 7.** WRF-GC model scalability by processes. Gray lines indicate perfect scalability, i.e. halved computational time for each doubling of processor cores.

**Figure S3.** Scalability test of the WRF-GC model on the Amazon Web Services using up to 64 nodes and 4,608 cores. The simulation domain was over the continental U.S. at 5 km × 5 km resolution (950 × 650 atmospheric columns), using 10-second dynamical time step and 5-minute external chemical time step.

ference in number of species advected.

**P16, L489-498, Section 5.1:**

A side-by-side wall time comparison between WRF-GC and WRF-Chem is difficult to do, because (1) the chemical routines in the two models are very different, and (2) the WRF-Chem has many possible configurations for chemistry. Nevertheless, we conducted one test simulation with WRF-Chem using a typical chemistry option (CBMZ-MOSAIC gas-phase chemistry, 4-bin aerosol microphysics and chemistry, and aqueous reactions; `chem_opt = 9` in `registry.chem`) with a total of 133 chemical species. This WRF-Chem simulation was configured for the same domain, at the same horizontal and vertical resolutions, and used the same physical and dynamical options as those in the WRF-GC simulation described above. The total wall time for the WRF-Chem simulation was 9985.8 seconds, which was almost twice as long as the wall time of WRF-GC (5127 seconds). Chemical routines in this WRF-Chem test simulation took up 61% of the total wall time (6134.3 seconds), despite the WRF-Chem having much fewer chemical species than the WRF-GC (241 chemical species). This may be partially due to the computationally intensive bin-resolved aerosol microphysics calculation in the WRF-Chem simulation.

*R2.29 Line 440: Change "light-weight" to "efficient".*

Thank you for your suggestion. We have revised the text:

**P17, L512-513, Section 5.2:**

However, the degradation had negligible impact on the total WRF-GC wall time as the WRF-GC Coupler was computationally efficient.

*R2.30 Line 449: The conclusion would benefit (perhaps at the end) about future directions. They are alluded to elsewhere in the text, but it would be good to briefly summarize them here. That would include things like more complex aerosol treatments and fully taking advantage of on-line coupling to include feedback effects (both of which are already in WRF-Chem). The conclusion would also be strengthened to foreshadow what would be discussed in part 2.*

Thank you for your suggestion. We revised the last paragraph to describe future plans for WRF-GC development.

**P18, L548-553, Section 6:**

The WRF-GC model is free and open-source to all users. The one-way coupled version of WRF-GC (v1.0) is now publicly available at `wrf.geos-chem.org`. A two-way coupled version with chemical feedbacks from GEOS-Chem to WRF is under development and will be presented in a forthcoming paper. Further development of WRF-GC will aim to enable nested-domain simulations, support size-dependent aerosol microphysical calculations, as well as further improve the physical compatibility with WRF. We envision WRF-GC to become a powerful tool for research, forecast, and regulatory applications of regional atmospheric chemistry and air quality.

*R2.31 Lines 460-463. As I mentioned earlier, the evaluation is rather simple. This paragraph should be rephrased to indicate this is only one case study (and thus not comprehensive) and other meteorological differences may be contributing to the PM2.5 predictions.*

Thank you for your suggestion. We have revised the text following your comment. Please also refer to R2.25, where we have added validation of other simulated meteorological fields against the observations.

**P17, L533-537**

Our first application showed that the WRF-GC model was able to reproduce the spatiotemporal variation of surface PM$_{2.5}$ concentrations over China in January 2015, with smaller biases compared to the results of the GEOS-Chem Classic nested-China simulation. This was partially because the WRF-GC model better represented the pollution meteorology, including the variability of the planetary boundary layer heights, over the region. In addition, the WRF-GC simulation was 3 times faster than a comparable GEOS-Chem Classic nested-grid simulation.

*R2.32 Line 464: The first sentence is an overstatement and deceiving if a reader only looks at the conclusions. Yes, the chemistry part itself scales well but not the entire code.*

Thank you for pointing this out. We have revised the text for accuracy. We also added a scalability analysis of the WRF-GC running on the Amazon Web Services.

**P17, L516-521, Section 5.2:**

We conducted a test simulation running WRF-GC on the Amazon Web Services (AWS) cloud with up to 128 nodes and 4608 cores. The simulation domain was over the continental U.S. at 5 km × 5 km resolution (950 × 650 atmospheric columns), with 10-second dynamical time step and 5-minute chemical time step. The scalability test results are shown in Figure S3. In this massively parallel environment, WRF-GC scaled well up to 1728 cores,

with the chemical module scaling well up to 2304 cores. The WRF-GC Coupler took less than 0.2% of the total computational time in this simulation and scaled perfectly up to 4608 cores.

**P17-18, L538-542, Section 6:**

WRF-GC demonstrated good scalability to massively parallel architectures, with near-perfect scalability of its chemistry component. This enables the WRF-GC model to be used on multiple-node systems or high-performance cloud computing platforms, which is not possible with the GEOS-Chem Classic. The GCHP model also scales to massively parallel architectures (Zhuang et al., 2020), but GCHP can only operate as a global model. The deployability of WRF-GC on the cloud will enhance WRF-GC's accessibility to new users.

**3    Executive Editor**

*R3.1 In particular, please note that for your paper, the following requirement has not been met in the Discussions paper:*

*- "The main paper must give the model name and version number (or other unique identifier) in the title."*

*Please add a version number WRF and GEOS-CHEM in the title upon your revised submission to GMD (as named in the code availability section).*

Thank you for your suggestion. We have revised the title of the paper:

[revised manuscript text omitted]